# ER procollagen storage defect without coupled unfolded protein response drives precocious arthritis

Kathryn M Yammine[1], Sophia Mirda Abularach[1], Seo-yeon Kim[1], Agata A Bikovtseva[1], Jinia Lilianty[2,3], Vincent L Butty[4], Richard P Schiavoni[5], John F Bateman[2,3], Shireen R Lamandé[2,3,6], Matthew D Shoulders[1,5,7]

Collagenopathies are a group of clinically diverse disorders caused by defects in collagen folding and secretion. For example, mutations in the gene encoding collagen type-II, the primary collagen in cartilage, can lead to diverse chondrodysplasias. One example is the Gly1170Ser substitution in procollagen-II, which causes precocious osteoarthritis. Here, we biochemically and mechanistically characterize an induced pluripotent stem cell-based cartilage model of this disease, including both hetero- and homozygous genotypes. We show that Gly1170Ser procollagen-II is notably slow to fold and secrete. Instead, procollagen-II accumulates intracellularly, consistent with an endoplasmic reticulum (ER) storage disorder. Likely owing to the unique features of the collagen triple helix, this accumulation is not recognized by the unfolded protein response. Gly1170Ser procollagen-II interacts to a greater extent than wild-type with specific ER proteostasis network components, consistent with its slow folding. These findings provide mechanistic elucidation into the etiology of this disease. Moreover, the easily expandable cartilage model will enable rapid testing of therapeutic strategies to restore proteostasis in the collagenopathies.

## Introduction

Collagens are the most abundant proteins in the human body and the molecular scaffold for animal life (Shoulders & Raines, 2009). Collagenopathies are a collection of genetic diseases caused by defects in the synthesis, folding, quality control, assembly, and/or function of members of the collagen protein family (Bateman et al, 2009). Autosomal dominant mutations in the >40 human genes that encode the 28 different types of collagen are the most common cause of collagenopathies, with mutations in 29 of these collagen genes already associated with disease (Hamosh et al, 2005; Jobling et al, 2014). As such, collagenopathies present with clinically diverse phenotypes that manifest most strongly in tissue. Currently, they have no cure. Moreover, effective treatments for most of the collagenopathies are nearly universally lacking (Bateman et al, 2009; Wong & Shoulders, 2019).

The fibrillar collagen type-II (collagen-II) is the main proteinaceous component of cartilage, and is thus a central player in both joint function and endochondral bone formation (Horton & Machado, 1988; Eyre, 2001). Mutations in the gene encoding collagen-II (*COL2A1*) often lead to diseases termed type-II collagenopathies, including many chondrodysplasias (Gregersen & Savarirayan, 1993; Spranger et al, 1994). These disorders encompass a broad range of clinical phenotypes, including skeletal dysplasia, ocular symptoms, and hearing impairment. The spectrum of severity ranges from relatively mild forms of premature arthritis to perinatal lethality. Type-II collagenopathies are overwhelmingly caused by mutations specifically within the procollagen-II triple helix-encoding portion of the *COL2A1* gene (UniProt Consortium, 2023). This >1,000 amino acid-long domain is composed of Xaa-Yaa-Gly repeats, and is bookended by two smaller globular domains that are cleaved post-secretion (Bauer & Lanschuetzer, 2003). Mutations that encode substitutions of the conserved Gly residues found at every third position in the triple-helical domain are the most common cause of disease (UniProt Consortium, 2023).

One exemplary type-II collagenopathy is the chondrodysplasia caused by the autosomal-dominant c.3508 GGT > AGT mutation in exon 50 of *COL2A1*, leading to a p.Gly1170Ser substitution in the C-terminal region of the procollagen-II triple-helical domain (Fig 1A) (Liu et al, 2005; Su et al, 2008). Patients harboring this mutation present with precocious osteoarthritis and Legg-Calvé-Perthes disease (a juvenile form of avascular necrosis of the femoral head), with symptoms reported as early as the first decade of life, and most often diagnosed in adulthood (Liu et al, 2005; Miyamoto et al, 2007; Su et al, 2008; Wang et al, 2014; Zhang et al, 2021). The disease is typically diagnosed via genetic screening engendered by the familial link. Treatment options are currently extremely limited, with total joint replacement as the standard-of-care.

---

[1]Department of Chemistry, Massachusetts Institute of Technology, Cambridge, MA, USA   [2]Murdoch Children's Research Institute, Parkville, Australia   [3]Department of Paediatrics, University of Melbourne, Parkville, Australia   [4]BioMicro Center, Massachusetts Institute of Technology, Cambridge, MA, USA   [5]Koch Institute for Integrative Cancer Research, Massachusetts Institute of Technology, Cambridge, MA, USA   [6]The Novo Nordisk Foundation Center for Stem Cell Medicine (reNEW), Murdoch Children's Research Institute, Parkville, Australia   [7]Broad Institute of MIT and Harvard, Cambridge, MA, USA

Correspondence: mshoulde@mit.edu

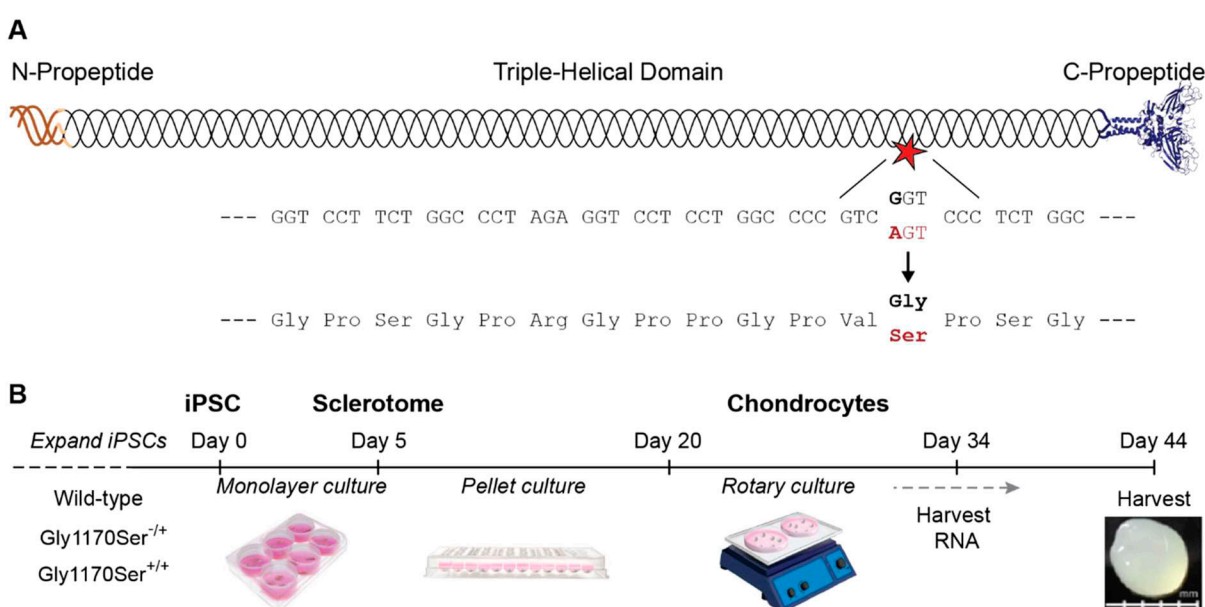

**Figure 1. Generating Gly1170Ser chondronoids.**
**(A)** Procollagen-II is composed of a lengthy (>1,000 amino acid) triple-helical domain sandwiched by two smaller propeptide domains. The c.3508 GGT > AGT mutation in exon 50 results in a Gly→Ser substitution at position 1170 in the procollagen-II amino acid sequence (shown in red). **(B)** Differentiation scheme from induced pluripotent stem cells to chondronoids.

A main challenge with studying collagenopathies in general, and type-II collagenopathies in particular, is the paucity of suitable systems to model these disorders. The disease phenotypes are most strongly manifested in tissue, making classical cell culture models insufficient. This challenge is especially acute for collagen-II and cartilage, as primary chondrocytes rapidly de-differentiate in typical culture conditions (Schnabel et al, 2002), and immortalized chondrocytes generally do not deposit an extensive extracellular matrix (ECM) (Goldring & Berenbaum, 1999). Moreover, mouse models of collagenopathies often fail to recapitulate human phenotypes, with heterozygous mice often showing little-to-no skeletal pathology, and homozygous mice presenting with much more severe pathology than that observed in patients.

Excitingly, recent advances in differentiating induced pluripotent stem cells (iPSC) towards chondrogenic lineages hold considerable promise for addressing this fundamental challenge, allowing us to begin to model type-II collagenopathies in vitro. Several protocols now exist for differentiating human pluripotent stem cells to the chondrogenic lineage through various growth factor treatments (Oldershaw et al, 2010; Koyama et al, 2013; Craft et al, 2015; Lee et al, 2015; Okada et al, 2015; Adkar et al, 2019; Lamandé et al, 2023). The inherent ability of stem cells to proliferate makes these systems highly expandable and thus much more amenable for biochemistry and drug discovery than mouse models. Methods for producing cartilage in vitro in the correct developmental context, combined with gene editing technology to create genetically matched cell lines incorporating disease-causing mutations (Hosseini Far et al, 2019; Howden et al, 2019; Kung et al, 2020; Lilianty et al, 2020, 2021; Yammine et al, 2023), provide an exceedingly promising avenue for modeling, studying, and discovering potential therapies for genetic skeletal disorders, including the type-II collagenopathies.

# Results

## Cells expressing Gly1170Ser-substituted collagen-II deposit a cartilaginous ECM

To elucidate the etiology of disease caused by the Gly1170Ser substitution in procollagen-II, in prior work we created and characterized genetically matched iPSC lines harboring the Gly1170Ser mutation in *COL2A1* in either one (Kung et al, 2020) or both alleles using CRISPR/Cas9 gene editing (Figs 1A and S1A–C and S2A and B and Table S1). These lines, along with the isogenic wild-type (WT) control, allow us to unambiguously assess the effect of the Gly1170Ser substitution on collagen-II proteostasis—meaning its synthesis, folding, localization, and function—and how it affects the cells that express it and the matrix they deposit. We began by differentiating all three iPSC lines (WT, heterozygous Gly1170Ser, and homozygous Gly1170Ser) to chondrocytes using the protocol recently described by Lamandé et al (2023). Briefly, we expanded iPSCs in feeder-free monolayer culture, then guided them through differentiation to sclerotome by growth factor treatment over the course of 4 d. To make individual spheroids, we transferred the cells to pellet culture for 2 wk and finally to swirler culture, which promotes chondrocyte differentiation and deposition of cartilage ECM components (Fig 1B) (Lamandé et al, 2023). We repeated experiments in three independent differentiations. After 44 d in culture, we harvested the chondronoids from all replicates and processed them for downstream assays.

To evaluate successful differentiation to chondrocytes, we interrogated differentiation-induced changes in the transcriptional profile of chondronoids after 44 d in culture using RNA-sequencing. The gene expression profile for all genotypes was consistent with

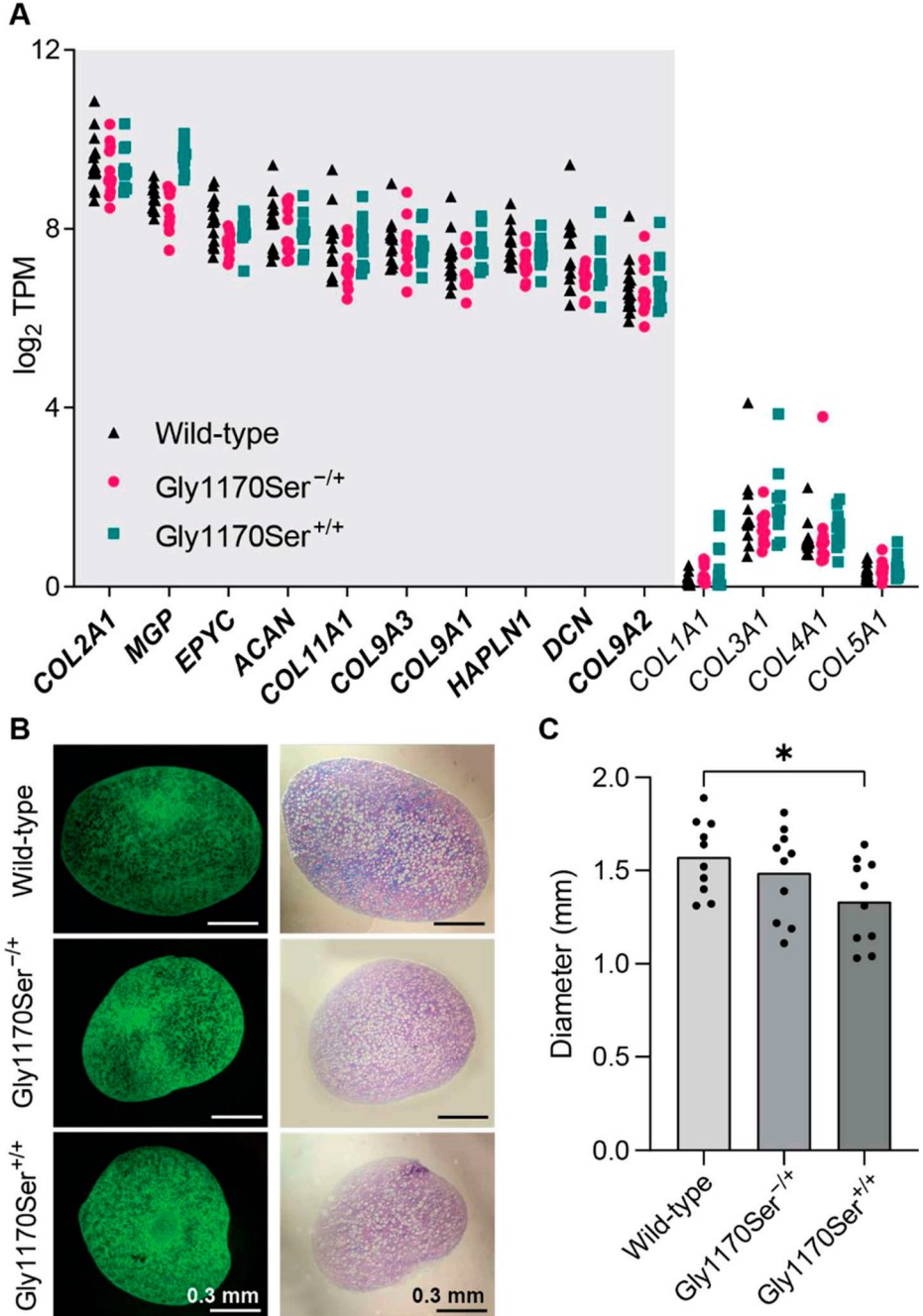

**Figure 2. Cells expressing Gly1170Ser procollagen-II deposit a cartilage matrix.**
**(A)** Ten most highly expressed core matrisome genes (bolded) demonstrating high expression of cartilage-specific markers. Low-to-no expression of collagens of other tissue types highlights the faithful differentiation to chondrocytes. RNA samples were harvested 44 d after initiation of the chondrocyte differentiation, from a total of 10 individual chondronoids per genotype across all three differentiations. TPM, transcripts per million. **(B)** Immunohistochemistry of collagen-II on histological sections of chondronoids shows that all genotypes deposit a matrix rich in collagen-II (green). Toluidine blue staining (purple) reveals an extensive proteoglycan matrix in all genotypes. **(C)** Size distribution of chondronoids by genotype. Homozygous Gly1170Ser chondronoids were significantly smaller than WT ($P < 0.05$).

that of chondrocytes, as evidenced by high expression of cartilage-specific markers, such as appropriate collagens (*COL2A1*, *COL11A1*, *COL9A1*) and aggrecan (*ACAN*), but not of collagens associated with other tissue types, including *COL1A1*, the major component of bone (Fig 2A). Indeed, cartilage components were the most highly expressed core matrisome genes (Hynes & Naba, 2012).

To assess the bulk composition of the ECM deposited by chondrocytes expressing the different procollagen-II variants, we first looked for the most abundant cartilage ECM components, collagen-II and proteoglycans (Eyre, 2001), in histology sections. We

observed that the WT, clinically relevant heterozygote, and the more severe homozygote mutant abundantly deposited collagen-II in the ECM (Fig 2B, in green). Similarly, toluidine blue staining for proteoglycans revealed that, consistent with the WT, the matrices of the heterozygous and homozygous Gly1170Ser variants were also rich in proteoglycans (Fig 2B, in purple). There was no significant difference in the size of the chondronoids between the WT and clinically relevant Gly1170Ser heterozygote. However, chondronoids expressing homozygous Gly1170Ser collagen-II were smaller than the WT controls (Fig 2C), hinting at a possible defect in secretion

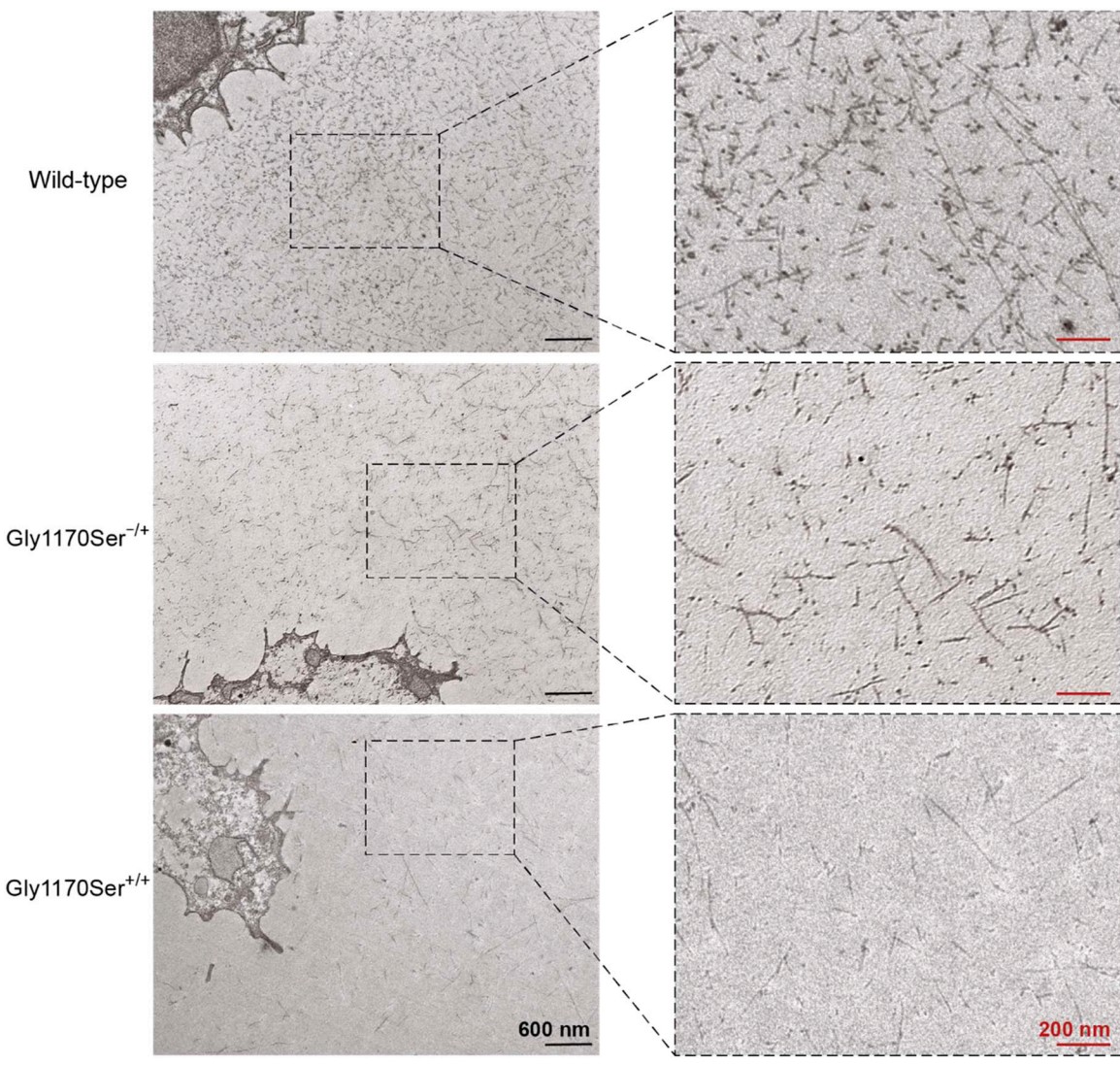

**Figure 3. Representative transmission electron microscopy images of chondronoid ECMs.**
Whereas the WT matrix is rich in network-forming collagen fibers, the clinically relevant heterozygous matrix is comparatively deficient, a phenotype that is further exacerbated in the homozygous chondronoids. At least three chondronoids were imaged per genotype.

of ECM components. Nevertheless, the abundant presence of collagen-II and proteoglycans indicates that, from a bulk composition perspective at least, chondrocytes expressing Gly1170Ser-subsituted procollagen-II deposit a cartilaginous ECM.

### The matrix of disease variant chondronoids is deficient

We hypothesized that, despite the deposition of an ECM rich in collagen-II and proteoglycans, and given the osteoarthritic and avascular necrotic phenotype of patients (Liu et al, 2005; Miyamoto et al, 2007; Su et al, 2008), expression of the disease-causing Gly1170Ser variant could lead to defects in the deposited ECM that were not apparent in the immunohistochemistry.

To assess the quality of the deposited matrix, we used transmission electron microscopy (TEM) to examine the chondronoid matrix at higher magnification and resolution. In the WT

chondronoids, we observed an extensive network of connecting fibrils (Fig 3), consistent with in vivo cartilage samples (Eyre, 2001). TEM imaging of the heterozygous Gly1170Ser chondronoids, in contrast, revealed a comparatively defective matrix (Fig 3). There were fewer collagen-II fibrils in the ECM, leading overall to a deficient network. This reduced matrix could be caused by insufficient or improperly folded collagen-II being secreted from cells and deposited in the ECM, by defective crosslinking of the collagen that is secreted into the ECM, or by a combination of both. This phenotype observed in the clinically relevant heterozygous Gly1170Ser variant could help to explain the joint symptoms of patients (Liu et al, 2005; Miyamoto et al, 2007; Su et al, 2008). Such a deficient matrix would likely lead to a defective coating of articular cartilage at joints, leaving the joint as a whole more susceptible to injury or the stress of repeated loading (Peters et al, 2018). When compared with the WT chondronoids, and even

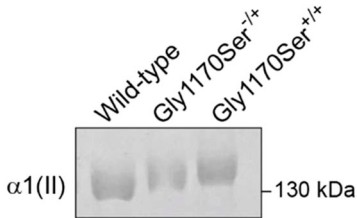

**Figure 4. SDS–PAGE separation of chondronoid-extracted collagen showing reduced electrophoretic mobility for the heterozygous and homozygous variants compared with WT collagen-II.**
4–5 chondronoids of each genotype were pooled for collagen extractions.

the heterozygotes, homozygous Gly1170Ser matrices showed striking deficiencies via TEM (Fig 3). The fibrils in chondronoids from this genotype were generally shorter, yielding a very sparse network.

### Gly1170Ser-substituted collagen-II is slow to fold, a defect that is exacerbated in homozygotes

Because expression of Gly1170Ser-substituted collagen-II leads to a deficient cartilage matrix, we next sought to elucidate the underlying cause(s) of this deficiency. We hypothesized that the Gly→Ser substitution causes a delay or defect in the proper folding and assembly of the procollagen-II triple helix (Bateman et al, 2009; Wong & Shoulders, 2019), which in turn could propagate to slow secretion of the protein into the ECM.

As procollagen-II folds in the ER, it interacts with modifying enzymes that install post-translational modifications on the individual polypeptide chains, such as hydroxyl groups on proline and lysine residues and *O*-glycans on hydroxylysine residues (Shoulders & Raines, 2009; Yamauchi & Sricholpech, 2012). Slower folding of the polypeptide chains leads to hyper accumulation of these modifications. Thus, a key signature of slow-folding collagen variants is reduced electrophoretic mobility on SDS–PAGE (Bateman et al, 1984, 1986; Bonadio et al, 1985; Godfrey & Hollister, 1988; Cabral et al, 2014), with *O*-glycosylation of the excessively hydroxylated lysine residues contributing to most of the shift in molecular weight.

To test the hypothesis that the Gly1170Ser pathologic substitution causes a delay in procollagen-II folding, we extracted collagen from the tissues by flash-freezing chondronoids and homogenizing the tissue, followed by a pepsin digestion to extract the pepsin-resistant collagen fraction. After separation via SDS–PAGE, we observed that the heterozygous, pathological Gly1170Ser substitution did indeed lead to reduced electrophoretic mobility, indicating that the substituted protein was excessively post-translationally modified and, therefore, slow to fold within cells when compared with the WT protein (Figs 4 and S3). The presence of a single broad, slow-migrating band as opposed to distinctive overmodified mutant versus normally modified WT strands is likely due to the fact that the majority of the trimers formed in heterozygotes (>85%) contain at least one Gly1170Ser strand that delays triple-helix folding.

In the homozygote, where all three procollagen-α1(II) chains contain the Gly1170Ser substitution, we observed that the delay in folding was further exacerbated (Fig 4). That is, the homozygous Gly1170Ser collagen-II displayed even further reduced electrophoretic mobility on SDS–PAGE compared with the heterozygous variant.

### Gly1170Ser-substituted procollagen-II is intracellularly retained and causes ER dilation

The deficient ECM observed, combined with the slow procollagen folding kinetics, strongly suggests that Gly1170Ser-substituted procollagen-II presents a challenge for cells to fold and secrete. We next sought to determine the fate of this variant in cells. We hypothesized that the slow folding of Gly1170Ser procollagen-II would lead to accumulation of the protein within cells and, specifically, within the ER, where this secreted protein folds. Intracellular retention is commonly observed in other diseases that impact protein folding and assembly (Teckman & Perlmutter, 2000).

To test this hypothesis, we visualized procollagen-II inside the chondrocytes. We observed that a significantly larger proportion of Gly1170Ser-expressing chondrocytes retained procollagen-II intracellularly as compared with the WT, as evidenced by collagen-II immunohistochemistry (IHC) (Fig 5A). We performed three independent, blinded scorings of multiple IHC images to quantify the fraction of cells with intracellularly retained procollagen-II in each genotype. In the clinically relevant heterozygous variant, 64% of cells showed signs of intracellular procollagen-II retention, as compared with just 34% of cells expressing WT procollagen-II (Fig 5B). In the more severe homozygous variant, >80% of cells exhibited intracellular procollagen-II retention (Fig 5B).

To better localize the intracellular protein, we co-stained with antibodies against collagen-II and an ER marker, calreticulin. We found that much of the intracellularly retained procollagen-II co-localized with calreticulin, suggesting that a portion of the protein is retained within the ER (Fig 5C). These observations, along with the slow folding kinetics (Fig 4), indicate that this pathologic variant is challenging for cells to properly fold and secrete, presumably because the sterically bulky residue in place of the much smaller Gly residue hinders triple helix assembly and disrupts a key stabilizing interchain hydrogen bond within the helix (Shoulders & Raines, 2009). Instead, procollagen-II begins to strongly accumulate within the ER.

We next sought to ascertain whether this intracellular retention dysregulates ER homeostasis, thereby exacerbating pathology. In other protein folding disorders, including other collagenopathies, intracellular retention of misfolded or aggregated protein is often accompanied by dilation of the ER (Teckman & Perlmutter, 2000). Indeed, high magnification TEM revealed that expression of the pathologic Gly1170Ser variant did indeed lead to dilated ER in the chondrocytes (Fig 6), a phenotype that was exacerbated in the homozygote. Consistent with ER storage diseases, this pathologic ER distention confirms that the Gly1170Ser substitution leads to defects in procollagen-II proteostasis, which in patients would eventually propagate to tissue-level defects in cartilage and pathologic joints.

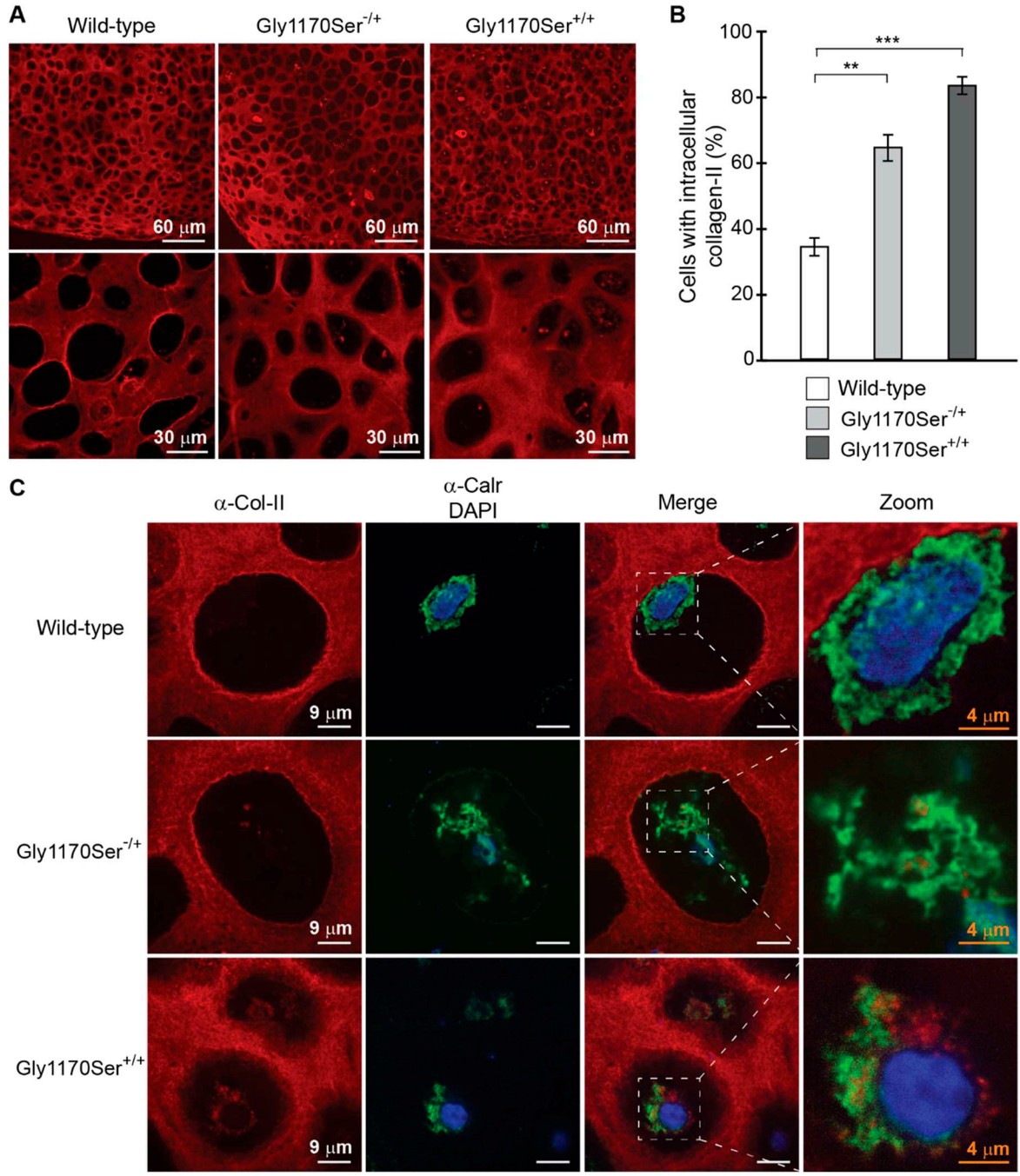

**Figure 5. Gly1170Ser procollagen-II is intracellularly retained.**
**(A)** Confocal images of collagen-II IHC at 10× and 63× magnification show an increase in intracellular procollagen-II retention across genotypes. **(B)** Blinded quantification of the extent of procollagen-II intracellular retention. 34% of WT cells have intracellular procollagen-II, 64% of heterozygous cells, and 83% of homozygous cells. **P < 0.01, ***P < 0.001. **(C)** Confocal images of chondronoids stained for collagen-II, the ER marker calreticulin, and the nuclear stain DAPI reveal that much of the intracellularly retained procollagen-II in disease variants is localized in the ER.

## Cells do not robustly respond at the transcript level to the Gly1170Ser-substituted collagen-II ER storage defect

Our data indicate that Gly1170Ser-substituted procollagen-II is slow to fold, accumulates in the ER, and exhibits a secretion defect. That secretion defect produces pathologic ER phenotypes, including extensive dilation. We next asked whether this protein folding challenge was accompanied by a transcriptional stress response. The unfolded protein response (UPR) is generally responsible for maintaining ER proteostasis by increasing levels of ER chaperones, quality control factors, and secretion machineries through signaling mediated by the transcription factors XBP1s and ATF6(f), and

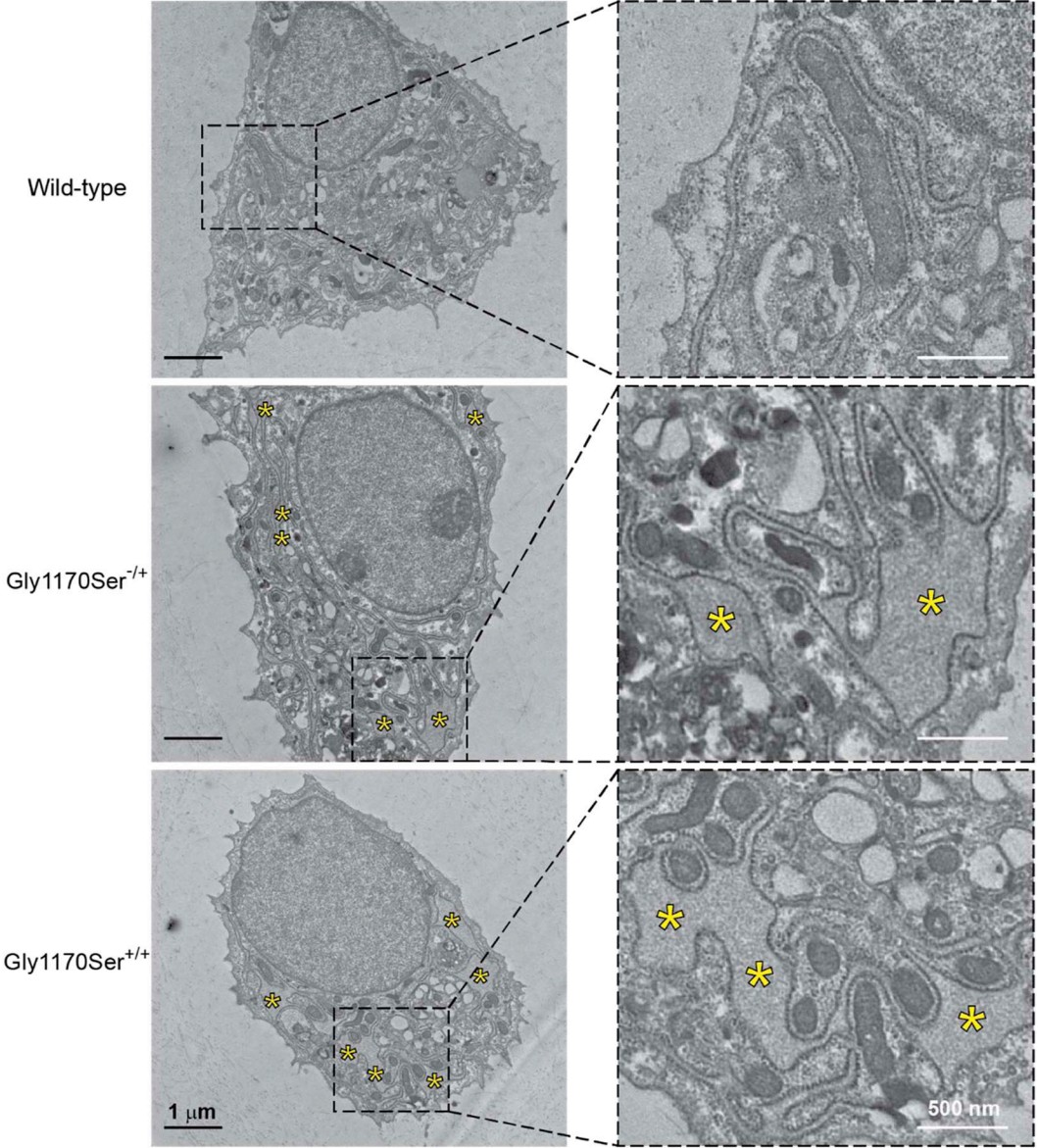

**Figure 6. Representative transmission electron microscopy images reveal the presence of distended ERs (marked with yellow stars) in cells expressing Gly1170Ser-substituted procollagen-II.**
The frequency and extent of ER dilation was further increased in homozygous cells compared with heterozygotes.

reducing non-essential translation via the PERK pathway (Walter & Ron, 2011). Upon sustained and unresolved UPR signaling, cellular apoptosis can also be induced by the PERK signaling pathway (Wong et al, 2018).

We tested for up-regulation of UPR target genes (Phillips et al, 2018) in Gly1170Ser-expressing chondronoids versus WT using RNA-sequencing of the different genotypes at two different timepoints during chondronoid growth: day 34 and day 44 (Fig 7). These timepoints were selected to reflect an early and a late stage of cartilage maturation, but with both timepoints harvested post-chondrogenesis so as not to interfere with the physiologic transient UPR activation that can be important in that process (Horiuchi et al, 2016). Notably, we observed that the UPR was not detectably

induced in the clinically relevant heterozygote line at either timepoint. For the more severe homozygote line, there was no sign of UPR activation at the later timepoint. At the early timepoint, there was a very modest up-regulation of UPR genes, which we discuss later in this section.

We next sought to determine if chondrocytes were instead raising any other organized transcriptional stress response when expressing Gly1170Ser procollagen. We looked for the non-canonical stress response to procollagen-I misfolding first described by Mirigian et al (2016) and Gorrell et al (2022), but we did not detect any significant up-regulation of genes associated with this response (Fig 7). We also scrutinized activation of the integrated stress response (ISR) (Grandjean et al, 2019; Gorrell et al, 2022) but

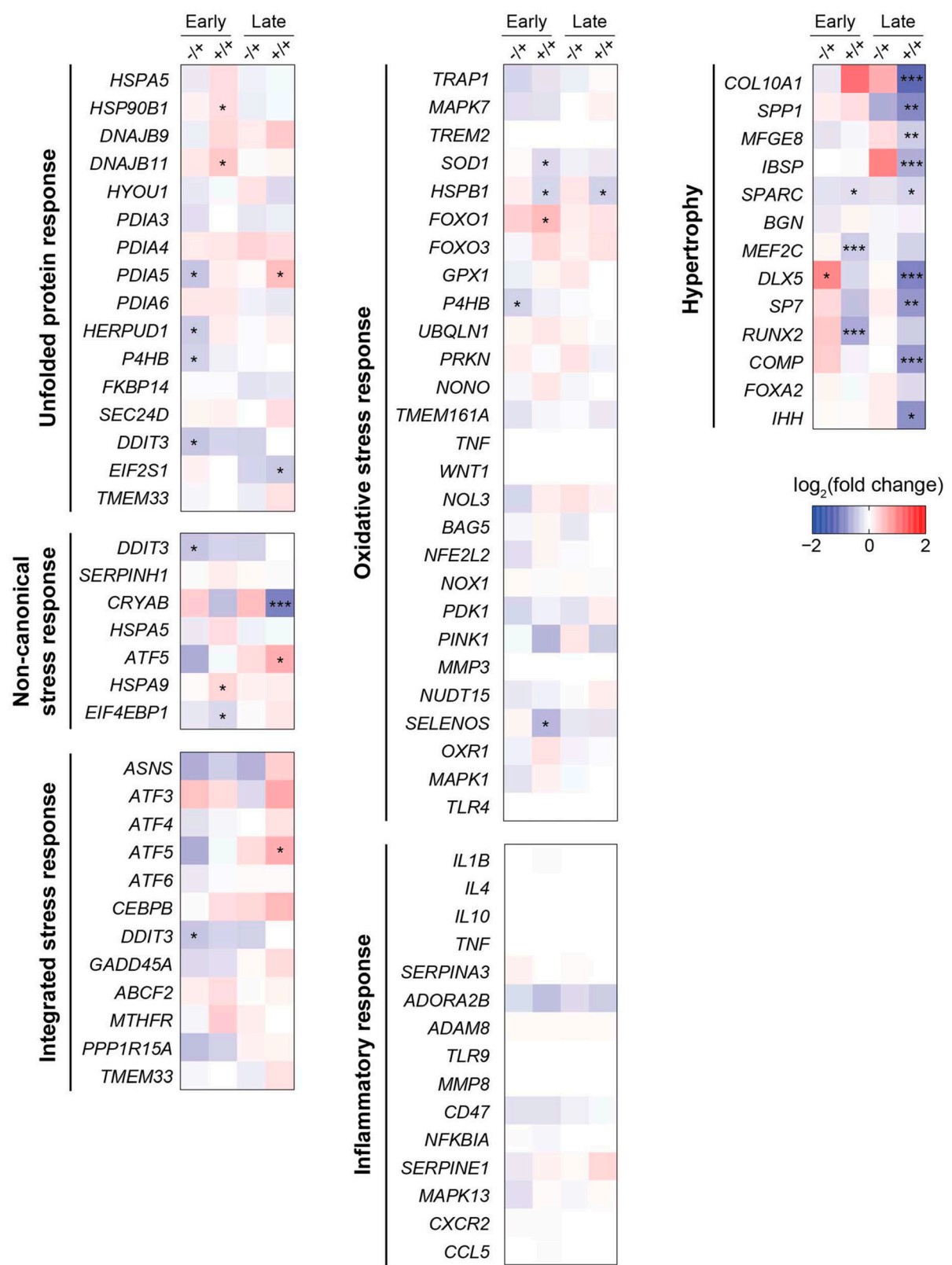

**Figure 7. Heatmap plotting the log$_2$(fold change) of selected genes expressed in the heterozygous (−/+) and homozygous (+/+) Gly1170Ser chondronoids compared with WT chondronoids, at the early and late time points.**

*p adjusted < 0.05; **p adjusted < 0.01; ***p adjusted < 0.001.

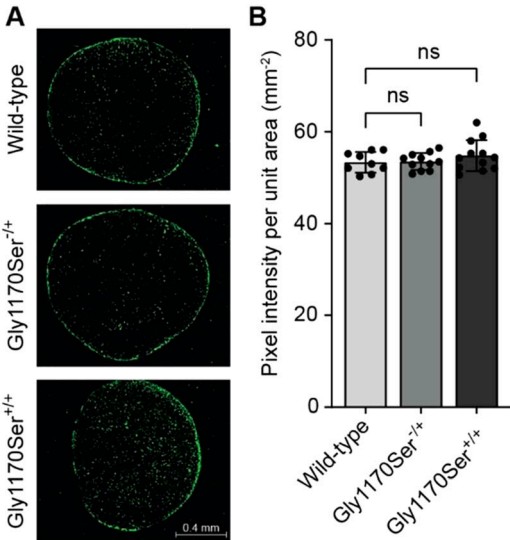

**Figure 8. TUNEL assay reveals no increase in apoptosis.**
**(A)** Representative images of TUNEL-stained chondronoid sections for each genotype. Breaks in DNA were labeled with fluorescent nucleotides (fluorescein, visualized in green). Apoptotic cells and residual sheared DNA appear TUNEL-positive. **(B)** Quantification of TUNEL signal (as pixel intensity per area) reveals no significant difference in apoptosis between genotypes.

observed no robust activation of this pathway for the clinically relevant heterozygote. There was, however, modest up-regulation of this response in the more severe but clinically irrelevant homozygote variant specifically at the later timepoint. Similarly, we investigated genes related to oxidative stress and inflammation, but again did not observe any significant differences in expression levels in the Gly1170Ser-expressing cells compared with WT, at either the early or late timepoints (Fig 7). These observations indicate that the chondrocytes were not raising such stress responses, at least when examined in the absence of paracrine signals from other cell types in the joint.

We next asked whether the challenge of folding Gly1170Ser procollagen-II in the ER causes apoptosis, despite no apparent induction of stress signaling at the transcriptional level. We performed a TUNEL assay to assess the number of apoptotic cells in chondronoids of the various genotypes. Consistent with our transcriptional data indicating a lack of any chronic UPR that could be pro-apoptotic, we did not observe an increase in apoptosis in the heterozygotes or the homozygotes (Figs 8A and B and S4).

We then turned to gene set enrichment analysis (GSEA) as a more global approach to identify differentially expressed pathways between Gly1170Ser-expressing lines and the WT (Figs S5 and S6). GSEA similarly did not identify any differentially expressed stress pathways. It did, however, highlight subtle differences in a few cellular processes. For the clinically relevant heterozygote, the main differences were up-regulation of gene sets related to cellular respiration, as well as a down-regulation of gene sets associated with translation and mitosis at the late timepoint when compared with the WT. These differences could suggest a UPR-independent reshaping of energetic allocations for cellular functions as an adaptive response to the accumulation of slow-folding procollagen-II. Moreover, the down-regulation of genes involved in

translation could be related to an attempt to alleviate the protein folding challenge in the ER, though the lack of apparent UPR/ISR activation makes it unclear how this response is initiated. Leiken and co-workers have described a non-canonical response (Mirigian et al, 2016; Gorrell et al, 2022), but as noted above that response is not observed in our transcription data.

On the other hand, for the homozygote line, GSEA revealed up-regulation of genes associated with collagen folding and ECM deposition (Fig S6). The implications of the observed differences in ECM-related processes will be interesting to follow-up on in future studies, albeit bearing in mind the reduced clinical relevance of this genotype. We also note that, possibly because cartilage-related pathways are generally poorly annotated, with few, if any, comprehensive gene sets, no cartilage-specific gene sets were identified as being differentially expressed in the Gly1170Ser-expressing cells. The most substantial transcriptional difference we observed, again present exclusively in the more severe and clinically irrelevant homozygous variant, was down-regulation of genes associated with chondrocyte hypertrophy (Lamandé et al, 2023), indicating that the homozygous chondrocytes were slower to mature when compared with WT (Fig 7). Chondrocyte hypertrophy, characterized by enlarged chondrocytes expressing collagen-X, is a key step in long bone formation, when the cartilaginous, collagen-II rich matrix shifts to a mineralized, collagen-I bone matrix in a process known as endochondral ossification (Horton & Machado, 1988). The up-regulation in the homozygous system of Matrix Gla Protein (Fig 2A), which inhibits vascular calcification of the matrix in vivo, further supports the delay in hypertrophy, and could lead to differences in the biomechanical properties of the matrix. Such slow maturation could also explain the very modest up-regulation of UPR targets observed exclusively at the early timepoint in the homozygous variant, as transient UPR signaling has been implicated in chondrocyte proliferation, differentiation, and hypertrophy (Wang et al, 2009; Horiuchi et al, 2016; Yuan et al, 2017).

Interestingly, the clinically relevant heterozygote showed the converse trend in hypertrophic gene expression, with certain genes for hypertrophy being up-regulated compared with the WT—a sign of accelerated maturation—although this effect was more muted (Fig 7). These differences in maturation speed likely have important implications for the overall functionality of the skeletal ECM deposited (Bateman et al, 2009). Further studies on the effect of maturation rate (specifically in the case of accelerated hypertrophy) are needed to better understand its role in pathogenesis of disease.

### Quantitative comparative interactome analysis of Gly1170Ser procollagen-II variant reveals increased interactions with certain ER enzymes and folding clients

Given the absence of robust, well-organized transcriptional responses, we next asked whether the cellular proteostasis machinery can recognize Gly1170Ser-induced procollagen-II misfolding. We used quantitative mass spectrometry (MS) to elucidate and compare the interactomes of WT versus Gly1170Ser-substituted procollagen-II (Fig S7) (DiChiara et al, 2016). For this experiment, we transiently overexpressed epitope-tagged procollagen-II in HT-1080 cells to enable MS-grade immunoprecipitations. This

cell line was selected because it expresses the cellular machinery necessary for folding procollagen, but does not express endogenous procollagen-II, which could confound our findings (Geddis & Prockop, 1993). Fig S7 outlines the experimental design and workflow.

Using tandem mass tag (TMT)-based MS, we quantitatively compared the interactomes of WT and Gly1170Ser procollagen-II to identify proteins that differentially interact with the bait (DiChiara et al, 2016; Doan et al, 2019, 2020; Plate et al, 2019). We identified multiple high confidence procollagen-II interactors (Table 1). After normalizing to bait, the fold interaction enrichment was calculated as

$$Fold\ Enrichment = \frac{\Sigma\ abundance\ in\ Gly1170Ser\ samples}{\Sigma\ abundance\ in\ wild-type\ samples}$$

The data reveal that the cellular proteostasis machinery does indeed differentially engage WT versus Gly1170Ser procollagen-II. Most notably, the pathogenic variant interacted to a greater extent with protein modifying and folding enzymes known to engage with procollagen in the ER (DiChiara et al, 2016). These interactors included procollagen-lysine,2-oxoglutarate 5-dioxygenase 2 (PLOD2), protein disulfide-isomerase (P4HB), prolyl 3-hydroxylase 1 (P3H1), peptidyl-prolyl cis–trans isomerase (FKBP10), and peptidyl-prolyl cis–trans isomerase B (PPIB) (Table 1). Heightened interactions with procollagen-modifying enzymes are consistent with the substituted variant being hypermodified owing to slow folding.

Other differentially interacting proteins included the known collagen chaperone calreticulin (CALR) and the collagen-specific molecular chaperone serpin H1 (SERPINH1) (Table 1). Increased interaction with these chaperones is consistent with Gly1170Ser procollagen-II being more challenging for cells to fold. Another notable interactor was the ER chaperone BiP (HSPA5) which interacted to the same extent with both WT and pathologic procollagen-II (presumably at their respective globular domains [Chessler & Byers, 1993; Lamandé et al, 1995; Doan et al, 2020]). This lack of differential interaction is consistent with the failure of Gly1170Ser procollagen-II to trigger the UPR, as seen in the RNA-sequencing data (Fig 7).

Beyond modifying enzymes and chaperones, other interactors that differentially engaged Gly1170Ser procollagen-II were also proteins destined for secretion, such as cartilage-associated protein (CRTAP) and fibronectin (FN1) (Table 1). These secreted proteins are, like procollagen-II, folded in the ER, where they apparently develop more extensive interactions with the slow-folding Gly1170Ser procollagen-II variant than with WT procollagen-II. Intriguingly, other disease-causing variants of procollagen-II were previously shown to have increased interactions with fibronectin, a finding suggested to underlie abnormal ECM formation (Ito et al, 2005).

Thus, the cellular proteostasis machinery can recognize slow-folding Gly1170Ser through increased interactions with certain ER proteostasis network components. These interactions do not alleviate the challenges associated with folding this variant, but they may present an opportunity for pharmacologic tuning of the ER proteostasis network to address the ER storage defect (Wong & Shoulders, 2019; Sebastian & Shoulders, 2020; Bateman et al, 2022).

## Discussion

Since it was first described in 2005 (Liu et al, 2005, 2018; Miyamoto et al, 2007; Su et al, 2008, 2010; Wang et al, 2014; Zhang et al, 2021), relatively limited progress has been made towards understanding the pathogenic mechanisms in the chondrodysplasia caused by the Gly1170Ser substitution in procollagen-II (Su et al, 2008; Liang et al, 2014; Yang et al, 2014; Lian et al, 2019). Here, we define the effects of the Gly1170Ser substitution in a robust human iPSC-based tissue model of cartilage and elucidate the proteostasis defects that arise from expression of this pathologic variant, in both the clinically relevant heterozygous model and the more severe but less clinically relevant homozygote. We find that, despite their ability to deposit a cartilage matrix rich in collagen-II and proteoglycans, cells expressing Gly1170Ser-substituted collagen-II do show signs of pathology, in both the hetero- and homozygous models. Their matrix is deficient and lacks much of the structural collagen network. At the cellular level, Gly1170Ser procollagen-II is slow to fold and accumulates excessive post-translational modifications. Delayed triple-helix folding also results in a large amount of procollagen being retained within a dilated ER. This ER accumulation is not, however, accompanied by any substantive UPR.

In a complementary approach, Liang et al previously generated a knock-in mouse model of the Gly1170Ser substitution, and characterized WT, heterozygous, and homozygous mice at birth or earlier (Liang et al, 2014). Heterozygous pups showed no skeletal phenotype, whereas homozygous pups presented with severe skeletal malformations and died shortly after birth from respiratory distress. Consistent with the findings from our human cartilage model, they observed a sparse ECM deposited by transgenic mice and ER dilation in both transgenic genotypes and intracellular retention of collagen-II, albeit only in chondrocytes homozygote for the substitution. Conversely, they also observed activation of the UPR in homozygous chondrocytes by quantitative RT–PCR of six UPR-related genes (and up-regulation of two of those genes in heterozygous chondrocytes). These differences highlight the intricacies of various model systems and underline the potential benefits of studying collagenopathies in models that most closely resemble the human context.

From a biophysical perspective, the pathologic Gly→Ser substitution disrupts the triple helix because it replaces the conserved, small glycine residue at the interior of the helix with a much larger serine, causing steric hindrance. Moreover, it disrupts a key interchain hydrogen bond that stabilizes the helix because the amide hydrogen of serine is not in the proper orientation to serve as a donor site (Shoulders & Raines, 2009). Such disruption slows triple-helix folding, causing the protein to accumulate post-translational modifications, which we detected by reduced electrophoretic mobility. The hypermodification may itself be problematic, as we will discuss later, but it is the slow folding leading to procollagen-II accumulation within the ER that is the hallmark of an ER storage disease (Rutishauser & Spiess, 2002).

Notably, even as Gly1170Ser procollagen-II accumulates in the ER, it does not trigger a chronic UPR. Bypassing the UPR despite extensive ER accumulation of a misfolding protein can occur if the accumulating protein evades detection by UPR signaling pathways.

**Table 1. High-confidence interactors[a] differentially engage with WT and Gly1170Ser procollagen-II across three biological replicates.**

| Protein (Gene) | Fold enrichment[b] | P-value[c] | Q-value[d] |
|---|---|---|---|
| Procollagen-lysine,2-oxoglutarate 5-dioxygenase 2 (PLOD2) | 1.95 | 0.0014 | 0.0181 |
| Protein disulfide-isomerase (P4HB) | 1.67 | 0.0004 | 0.0101 |
| Prolyl 3-hydroxylase 1 (P3H1) | 1.55 | 0.0352 | 0.1486 |
| Cartilage-associated protein (CRTAP) | 1.53 | 0.0328 | 0.1486 |
| Fibronectin (FN1) | 1.48 | 0.0397 | 0.1486 |
| Peptidyl-prolyl cis–trans isomerase FKBP10 (FKBP10) | 1.45 | 0.0450 | 0.1486 |
| Serpin H1 (SERPINH1) | 1.45 | 0.1294 | 0.2847 |
| Peptidyl-prolyl cis–trans isomerase B (PPIB) | 1.41 | 0.0324 | 0.1486 |
| Prolyl 4-hydroxylase subunit alpha-1 (P4HA1) | 1.41 | 0.0947 | 0.2777 |
| Protein canopy homolog 2 (CNPY2) | 1.36 | 0.3609 | 0.4793 |
| Peptidyl-prolyl cis–trans isomerase FKBP9 (FKBP9) | 1.36 | 0.1174 | 0.2847 |
| Glucosidase 2 subunit beta (PRKCSH) | 1.32 | 0.1252 | 0.2847 |
| Prolyl 4-hydroxylase subunit alpha-2 (P4HA2) | 1.31 | 0.1772 | 0.3340 |
| Calumenin (CALU) | 1.31 | 0.1904 | 0.3352 |
| Procollagen-lysine,2-oxoglutarate 5-dioxygenase 1 (PLOD1) | 1.29 | 0.2534 | 0.3935 |
| Calreticulin (CALR) | 1.28 | 0.0116 | 0.1021 |
| Multifunctional procollagen lysine hydroxylase and glycosyltransferase LH3 (PLOD3) | 1.24 | 0.3631 | 0.4793 |
| Procollagen galactosyltransferase 1 (COLGALT1) | 1.24 | 0.3873 | 0.4869 |
| Cytoskeleton-associated protein 4 (CKAP4) | 1.17 | 0.4509 | 0.5411 |
| Thioredoxin domain-containing protein 5 (TXNDC5) | 1.14 | 0.6372 | 0.6887 |
| Endoplasmic reticulum chaperone BiP (HSPA5) | 1.13 | 0.1698 | 0.3340 |
| Protein disulfide-isomerase A3 (PDIA3) | 1.13 | 0.2887 | 0.4233 |
| Endoplasmin (HSP90B1) | 1.13 | 0.2434 | 0.3935 |
| Protein ERGIC-53 (LMAN1) | 1.11 | 0.6354 | 0.6887 |
| Protein disulfide-isomerase A4 (PDIA4) | 1.09 | 0.6521 | 0.6887 |
| Protein disulfide-isomerase A6 (PDIA6) | 1.01 | 0.9746 | 0.9896 |

[a]Proteins displaying a > twofold increase in signal compared with negative control conditions in at least four the six experimental samples. See MassIVE dataset MSV000093517 for complete, unfiltered data set.
[b]Fold-enrichment was calculated as the ratio of bait-normalized peptides detected in Gly1170Ser to the bait-normalized peptides detected in WT.
[c]P-value was calculated using a homoscedastic $t$ test across the three biological replicates.
[d]Q-value was calculated using a two-stage step-up method of Benjamini, Krieger, and Yekutieli with a desired false discovery rate set to 10%.

The primary mechanism to activate the UPR relies on the ER chaperone BiP. Downstream UPR signaling is triggered when BiP is sequestered off the three UPR sensors, IRE1, ATF6 and PERK, to instead bind un- or misfolded proteins through hydrophobic interactions with exposed hydrophobic cores. Procollagen, however, is a long hydrophilic protein lacking such a core. Moreover, its triple helix—where the Gly1170Ser substitution is located—is adorned with many hydrophilic post-translational modifications (Eyre, 1987; Yamauchi & Sricholpech, 2012). In fact, there is currently no evidence that BiP binds the collagen triple helix: BiP does bind to procollagen's C-propeptide domain during folding, and interacts particularly strongly with misfolding-prone C-propeptide domain variants (Chessler & Byers, 1993; Lamandé et al, 1995; Wong & Shoulders, 2019; Doan et al, 2020; Li et al, 2021), but, again, there

is no evidence for BiP binding to the triple-helical domain of even disease-causing triple-helical domain variants of collagen-I.

Our data suggest that an underlying cause of pathology is the slow folding and accumulation of Gly1170Ser procollagen-II intracellularly, which occurs in the absence of any UPR or other stress response. Indeed, for many of the common Gly→Ser substitutions that induce collagenopathies, UPR activation is, at most, inconsistently observed (Bateman et al, 2022). Our results add weight to the view that chronic, pathologic UPR is by no means a universal mechanism underpinning the collagenopathies (Wong & Shoulders, 2019; Bateman et al, 2022). Relatedly, we do not detect any significant increase in apoptosis in cells expressing Gly1170Ser procollagen-II. This finding could be a product of the timepoint considered, but it is at least fully consistent with the

absence of chronic activation of the PERK arm of the UPR that can induce apoptosis over time.

From a therapeutic perspective, our finding that cells expressing the pathologic variant do not activate beneficial aspects of the UPR makes those pathways a compelling target for this, and likely many other, collagenopathies (Balch et al, 2008; Wong & Shoulders, 2019; Sebastian & Shoulders, 2020). Exogenous, branch-specific activation of the pro-folding IRE1 and ATF6 arms of the UPR could promote procollagen folding by up-regulating chaperones and quality control factors, as well as clearing accumulated protein by up-regulation of clearance machinery (Ryno et al, 2013; Shoulders et al, 2013). Even just one of these outcomes could greatly benefit cells expressing Gly1170Ser procollagen-II. This strategy has already been applied in other protein folding diseases (Ryno et al, 2013), and many small molecules are already known to specifically target different nodes within the UPR (Grandjean & Wiseman, 2020).

Further evidence that promoting proteostasis, whether via the UPR or more generally, may be a viable therapeutic strategy for this disease, derives from the location of the substitution within the collagen-II triple helix. Position 1170 does not lie in any known binding site for other proteins (Hamaia et al, 2012; Howes et al, 2014; Chen & Lin, 2019) and, therefore, likely does not impede important signaling or structural functions mediated by its interactors. Rather, it seems likely that pathology is due directly to proteostasis defects.

Generally, the nature and location of the Gly1170Ser substitution would suggest a much more severe pathology than that observed in both patients and in our cartilage model because (1) it involves a Gly substitution, which are generally more detrimental to procollagen folding than substitutions of Xaa- or Yaa-position residues (Bateman et al, 2009), and (2) it is C-terminal, and thus, because procollagen folds from the C- to N-terminus, would disrupt the folding of a much larger portion of the protein than if it were located near the N-terminus (Bodian et al, 2008). The symptoms of this disease, however, fall on the milder side of the spectrum, which is likely attributable to the proline-rich region located directly N-terminal to the substitution. The propensity of peptides to assemble into triple helices increases with the number of prolines (Shoulders & Raines, 2009) and, therefore, the proline-rich region likely helps procollagen-II to more rapidly resume folding despite the presence of a large, interchain hydrogen bond-breaking amino acid.

Despite its "favorable" position, Gly1170Ser still causes pathology. The ER storage phenotype associated with expression of Gly1170Ser procollagen-II propagates to the tissue, as these cells deposit a deficient ECM, lacking many of the network-forming fibrils observed in the WT. In patients, such failure likely leads to cartilage degradation ultimately requiring total joint replacement. From a molecular perspective, multiple factors could be causing the deficient matrix that ultimately contributes to symptoms in patients. One possibility is that cells expressing the pathologic variant secrete less collagen into the extracellular space. Another is that the secreted collagen is structurally abnormal and unable to correctly assemble into fibrils. Alternatively, or in conjunction, the deficient matrix could be a product of a defect in crosslinking. Hypermodification of lysines in collagen is known to affect crosslinking, as the process requires both a lysine and a hydroxylysine devoid of

O-glycans to form a covalent crosslink (Eyre, 1987). Our data indicate that Gly1170Ser-substituted procollagen-II is hypermodified because of slow folding, which likely leads to a higher proportion of O-glycosylated hydroxylysines, and therefore could disrupt crosslinking and overall matrix health.

Interestingly, whereas the slow folding of Gly1170Ser procollagen-II, intracellular accumulation of the protein, and deficient ECM phenotypes all display a stepwise difference in severity between the heterozygous and homozygous variants, our RNA-sequencing data reveal that the homozygous cells are not, at least from a transcriptomic standpoint, simply a more severe version of the heterozygotes. Both heatmaps and GSEA analyses revealed that different genes and pathways were differentially expressed depending on the genotype. This finding has implications for mouse models where homozygotes are often used to model autosomal dominant disorders in humans, owing to a lack of disease phenotype in mice heterozygous for the mutation. Nonetheless, our finding that homozygous chondronoids displayed a delay in hypertrophy is consistent with the severe skeletal dysplasia and disorganized growth plate observed in homozygous Gly1170Ser mice (Liang et al, 2014).

In this work, we focused on modeling growth plate cartilage to understand how the mutation impacts development of cartilage and the skeleton, and how those relate to pathology, in both the clinically relevant heterozygous variant and the more severe homozygote. In future work, it will be equally interesting to investigate the phenotype in articular cartilage, which can also be grown from iPSCs (Craft et al, 2015), to further understand how the mutation relates to the osteoarthritic phenotype in particular.

The comparison in a human model with isogenic background enabled by the chondronoid system allowed us to define the pathologic phenotype and elucidate the defects in procollagen-II proteostasis. Moreover, by virtue of being highly expandable and modeling tissue, chondronoids constitute an ideal platform for screening small molecules and genetic interventions to enhance procollagen-II proteostasis. The model is equally conducive to in depth biochemical analyses and, with recent advances in iPSC-based modeling of the endochondral ossification process (Lamandé et al, 2023), could enable similar studies in collagenopathies involving bone disorders.

# Materials and Methods

### General reagents

All media and cell culture reagents were obtained from Corning/Cellgro, unless otherwise noted. Restriction enzymes, ligases, and polymerases were obtained from New England BioLabs. HA-Agarose beads (A2095) were obtained from Millipore.

### Plasmids

The plasmid encoding WT human *COL2A1* (variant IIB, consensus sequence) was obtained from Origene (#RG221644; NM_033150).

The C-terminal GFP tag was removed and replaced by a stop codon. To the N-terminal side of the N-propeptide, a pre-protrypsin signal sequence, HA epitope tag, and NotI site were inserted. The Gly1170Ser *COL2A1* plasmid was made using site-directed mutagenesis of the WT plasmid.

## Antibodies

Antibodies were obtained from the following suppliers, and used at the specified dilutions: anti-Collagen-II (MAB8887, 1:150; Millipore), anti-Calreticulin (AB325990, 1:200; Invitrogen), Goat Anti-Rabbit Alexa Fluor 488 (ab150077, 1:500; Abcam), Goat Anti-Mouse Alexa Fluor 568 (ab175473, 1:500; Abcam).

## iPSC lines growth and maintenance

Generating the gene edited lines from ATCC fibroblast line 1,502 was covered by the Human Research Ethics Committee of the Royal Children's Hospital (HREC #33118) (Kung et al, 2020). iPSC lines MCRIi019-A (WT control) and MCRIi019-A-2 (*COL2A1* p.Gly1170Ser heterozygote) and the homozygote variant iPSCs were grown in a feeder-free manner on Matrigel (Corning)-coated six-well plates in Essential 8 (E8) medium (Thermo Fisher Scientific). Media was changed daily. Cells were routinely passaged (1:6 split) every 3–4 d and gently detached from plates using 0.5 mM EDTA in PBS.

## Chondrocyte differentiation

Undifferentiated iPSCs at 70–90% confluency were dissociated using 0.5 mM EDTA in PBS and passaged into six-well plates (~2 × 10$^5$ cells per well) pre-coated with Matrigel, and cultured for 48 h in E8 expansion medium. Differentiation to sclerotome was performed as described (Lamandé et al, 2023), with APEL2 (StemCell Technologies) containing 5% Protein-Free Hybridoma Medium (PFHM II; Thermo Fisher Scientific). On day 0, medium was changed to anterior primitive streak-inducing medium containing 30 ng/ml Activin A (R&D Systems), 4 $\mu$M CHIR99021 (Tocris), 20 ng/ml FGF2 (PeproTech) and 100 nM PIK90 (Merck Millipore). After 24 h, this medium was replaced with medium containing 3 $\mu$M CHIR99021, 20 ng/ml FGF2, 1 $\mu$M A8301 (Tocris) and 0.25 $\mu$M LDN193189 (Cayman Chemical) to induce paraxial mesoderm for 24 h. Early somite development was then induced with medium containing 1 $\mu$M A8301 and 0.25 $\mu$M LDN193189, 3 $\mu$M C59 (Tocris) and 0.5 $\mu$M PD0325901 (Selleck Chemicals). After 24 h, sclerotome induction was initiated with 1 $\mu$M C59 and 2 $\mu$M purmorphamine (Sigma-Aldrich). After the 1$^{st}$ d of sclerotome induction (day 4) cells were dissociated from monolayer culture using Tryple Select (Thermo Fisher Scientific), resuspended in sclerotome differentiation media, 300 $\mu$l aliquots containing 2 × 10$^5$ cells were dispensed into 96-well low-attachment, round-bottom plates (Corning), and pelleted by centrifugation at 400$g$ for 3 min using a swing-out rotor. Pellets were incubated in sclerotome differentiation media for a further 48 h to complete sclerotome differentiation in the 96-well static culture format. Experiments were performed in three independent differentiations, where individual replicates originated from cells of separate plates differentiated with individually prepared differentiation media from different media bottles.

## Chondrocyte maturation and ECM deposition

To promote chondrocyte differentiation, maturation, and ECM deposition, sclerotome cell pellets were cultured in 96-well round bottom plates in APEL2 medium supplemented with 5% PFHM II, 20 ng/ml FGF2 (PeproTech), 1× Penicillin-Streptomycin (Thermo Fisher Scientific), and 200 $\mu$M freshly added sodium ascorbate (Amresco). On day 20, pellets were transferred to 6 cm non-adherent culture dishes (Greiner) with 15–20 pellets per dish in 5 ml of APEL2/PFHM II medium lacking FGF2 with orbital rotation at 60 rpm on a LSE Low Speed Orbital Shaker (Corning). Media were changed every 2–3 d for an additional 24 d of maturation. For harvesting, chondronoids were rinsed once in PBS and snap-frozen or processed for downstream assays. For each assay, chondronoids were harvested from all three biological replicates. Chondronoids originating from the same differentiation were considered technical replicates. A micrometer was used for precise diameter measurement of each of 10 chondronoids per genotype.

## Histology and toluidine blue staining

Cultured chondronoids were fixed overnight at 4°C in 10% neutral-buffered formalin (Millipore), washed in 70% ethanol, processed on a tissue processor (Sakura Tissue Tek VIP5) with a biopsy processing schedule as follows: 70% ethanol—10 min, 95% ethanol × 2–10 min each, 100% ethanol × 3–10 min each, xylene × 2–10 min each, paraffin wax × 4–10 min each, and then embedded in "Paraplast X-tra" paraffin wax using a Leica embedding station. Serial 4 $\mu$m sections were cut from the chondronoids using a Leica rotary microtome. Sections were floated onto a 45°C water bath and mounted on Leica "X-tra" adhesive slides (Leica biosytems), drained, and dried at RT overnight. Before staining, sections were heated at 60°C for 30 min. Sections were treated with xylene to remove the paraffin wax and then with an ethanol series comprising 100%, 90%, 70% ethanol then water to rehydrate the sections. Sections were stained with toluidine blue solution (1% toluidine blue, 0.2 M sodium phosphate pH 9, filtered) for 10 min to detect a cartilage proteoglycan matrix, then rinsed in distilled water. Microscope slides were cover-slipped using ProLong Gold Antifade Reagent (Invitrogen) and sealed with nail polish. Images were captured using a Leica DM 2000 LED microscope with Leica Application Suite software version 4.9.0. Image processing was performed using ImageJ.

## Immunohistochemistry and confocal microscopy

Rehydrated histological sections underwent antigen retrieval by incubation with 1 mg/ml pepsin in 0.5 M acetic acid (Millipore) at 37°C for 30 min. Samples were washed 3× with PBS, then incubated with 0.1% Triton X-100 for 30 min to permeabilize the cells. Samples were then incubated in a solution of 5% BSA (Millipore) in PBS at RT for 1 h to block non-specific antibody binding. Samples were labeled simultaneously with anti-Col-II antibody and anti-Calr antibody in 1% BSA for 1 h at RT. Samples were washed 3 × with PBS. Both secondary antibodies, Alexa Fluor 568-conjugated anti-mouse and Alexa Fluor 488-conjugated anti-rabbit (1:500; Invitrogen), were applied simultaneously in 1% BSA solution for 30 min—1 h at RT.

Samples were washed 3× with PBS. Cover slips were mounted on the microscope slides using ProLong Gold Antifade Reagent with DAPI (Invitrogen) and sealed with nail polish. Images were acquired at the W. M. Keck Microscopy Facility at the Whitehead Institute on a Zeiss AxioVert200M microscope with a 10×, 63×, and 100× oil immersion objective with a Yokogawa CSU-22 spinning disk confocal head with a Borealis modification (Spectral Applied Research/Andor) and a Hamamatsu ORCA-ER CCD camera. The MetaMorph software package (Molecular Devices) was used for image acquisition. The excitation lasers used to capture the images were 405, 488, and 561 nm. Image processing was performed using ImageJ.

### TUNEL assay

Rehydrated histological sections were assayed for apoptosis using an in situ cell death detection kit (Millipore), according to the manufacturer's instructions. Five different chondronoids per genotype were used for analysis. Briefly, after cell permeabilization, DNA breaks were labeled directly by the enzyme terminal deoxynucleotidyl transferase (TdT) with fluorescein labeled dUTPs. Positive control sample was incubated with DNAse for 15 min at 37°C before TdT labeling. Negative control sample did not include the TdT enzyme. Samples were visualized on a fluorescence microscope (ECHO Revolve 4). Signal quantification was performed on the ECHO Pro App by analyzing the pixel intensity over the highlighted region of interest. Significance was assessed using a $t$ test on at least 10 different sections per genotype.

### TEM

Samples were washed with PBS and fixed with 2.5% glutaraldehyde, 2.0% PFA in 100 mM cacodylate buffer pH 7.2 for 2 h at 4°C, then post-fixed with 1% osmium tetroxide in 1.25% potassium ferrocyanide. They were then en-bloc stained with 2% uranyl acetate in 0.05 M maleate buffer pH 5.2 overnight at RT, followed by serial dehydrations with ethanol, then embedded in resin at 60°C for 48 h. 60 nm sections were obtained using a diamond knife on a Leica UC67 Ultramicrotome and observed at 120 kV on a T12 Spirit Transmission Electron Microscope (Thermo Fisher Instruments). Micrographs were captured using a digital camera from Advanced Microscopy Techniques.

### Collagen extraction and SDS–PAGE analysis

Chondronoids of the same genotype were rinsed with PBS, blotted dry with a Kimwipe, pooled, snap frozen in liquid nitrogen, and pulverized using a liquid nitrogen-cooled tissue grinder. The powder obtained was added to 100 $\mu$l chondroitinase ABC solution (50 mM Tris–HCl pH 8, 50 mM sodium acetate, 0.1 mU/$\mu$l chondroitinase ABC [Millipore]) and incubated in a thermomixer set to 37°C and 1,500 rpm for 6 h. 900 $\mu$l of denaturing solution (50 mM sodium acetate, 10 mM EDTA, 4 M guanidine hydrochloride, 65 mM DTT) was added, and samples were rotated end-over-end at 4°C overnight. The next day, samples were centrifuged at 16,000$g$ at 4°C for 10 min. The supernatant was removed and the pellet resuspended in 200 $\mu$l of cold pepsin solution (0.1 mg/ml pepsin [Millipore] in 0.5 M acetic acid) and rotated end-over-end at 4°C

overnight. The next day, samples were centrifuged at 16,000$g$ at 4°C for 10 min. The supernatant was saved, neutralized with 10 M NaOH, and lyophilized. Lyophilized samples were reconstituted in loading buffer for SDS–PAGE (50 mM Tris–HCl pH 6.8, 2 M urea, 0.1% SDS, 0.1% sucrose, 0.01% bromophenol blue [Millipore]) and boiled for 15 min. Samples were separated on homemade 4/8% Tris SDS–PAGE gels. Gels were stained in Coomassie Blue solution (0.1% Coomassie Blue [Millipore], 10% acetic acid, 50% methanol, 40% water) and destained in the same solution without Coomassie Blue. Gels were imaged on an Epson scanner.

### RNA-sequencing

For RNA-sequencing, individual chondronoids were snap frozen in liquid nitrogen and pulverized using a liquid nitrogen-cooled tissue grinder. RNA was extracted with TRIzol (Invitrogen), followed by purification using Direct-zol RNA Microprep kit spin columns (Zymo Research), according to the manufacturer's instructions. RNA samples were run on a fraction analyzer at MIT's BioMicro Center for quality control. All RNA samples were processed at the same time using the same batch of reagents. RNA samples were quantified using an Advanced Analytical Fragment Analyzer. 10 ng of total RNA was used for library preparation on a Tecan ECO150. Custom primers were used to append a 6 bp barcode specific to each sample and a unique molecular identifier (Integrated DNA technologies). Maxima H Minus Reverse Transcriptase (Thermo Fisher Scientific) was added per the manufacturer's recommendations with a template-switching oligo, incubated at 42°C for 90 min, and inactivated by incubation at 80°C for 10 min. After the template switching reaction, cDNA from 24 wells containing unique well identifiers were pooled together and cleaned using RNA Ampure beads at 1.0×. cDNA was eluted with 17 $\mu$l of water, digested with Exonuclease-I (New England Biolabs) at 37°C for 30 min, and inactivated by incubation at 80°C for 20 min. Second strand synthesis and PCR amplification was performed by adding Advantage 2 Polymerase Mix (Clontech) and the SINGV6 primer (Integrated DNA Technologies, 10 pmol, 5'-/5Biosg/ACACTCTTTCCCTACACGACGC-3') directly to half of the exonuclease reaction volume. Eight cycles of PCR were performed followed by clean up using regular SPRI beads at 0.6×, and elution with 20 $\mu$l of RSB. Successful amplification of cDNA was confirmed using the Fragment Analyzer. Illumina libraries were then produced using Nextera FLEX tagmentation substituting P5NEXTPT5-bmc primer 25 $\mu$M, Integrated DNA Technologies, (5'-AATGATACGGCGACCACCGA-GATCTACACTCTTTCCCTACACGACGCTCTTCCG*A*T*C*T*-3' where * = phosphorothioate bonds.) in place of the normal N500 primer. Final libraries were cleaned using SPRI beads at 0.7× and quantified using the Fragment Analyzer and qRT-PCR before being loaded for paired-end sequencing using an Illumina NextSeq500 in high-output paired-end mode (20/57 nt reads). Analyses were performed using previously described tools and methods. Reads were aligned against hg19 (February 2009) using BWA mem v.0.7.12-r1039 (RRID:SCR_010910) with flags −t 16 −f, and mapping rates, fraction of multiply-mapping reads, number of unique 20-mers at the 5' end of the reads, insert size distributions and fraction of ribosomal RNAs were calculated using BEDTools v. 2.25.0 (RRID:SCR_006646) (Quinlan & Hall, 2010). In addition, each

resulting bam file was randomly down-sampled to a million reads, which were aligned against hg19, and read density across genomic features were estimated for RNA-Seq–specific quality control metrics. For mapping and quantitation, reads were scored against GRCh38/ENSEMBL 101 annotation using Salmon v.1.3 with flags quant-p 8-l ISR –validateMappings (Patro et al, 2017). The resulting quant.sf files were imported into the R statistical environment using the tximport library (tximport function, option "salmon"), and gene-level counts and transcript per milllion estimates were calculated for all genes (Soneson et al, 2015). Samples were clustered based on $\log_2$-transformed transcript per milllions of protein-coding genes and obvious outliers were excluded from downstream analyses.

Differential expression was analyzed in the R statistical environment (R v.3.5.1) using Bioconductor's DESeq2 package (RRID: SCR_000154) (Love et al, 2014). Dataset parameters were estimated using the estimateSizeFactors(), and estimateDispersions() functions; read counts across conditions were modeled based on a negative binomial distribution, and a Wald test was used to test for differential expression (nbinomWaldtest(), all packaged into the DESeq() function), using the treatment type as a contrast. Shrunken $\log_2$ fold-changes were calculated using the lfcShrink function, based on a normal shrinkage estimator (Love et al, 2014). Fold-changes, P-values and false-discovery rates were reported for each gene. In the heterozygous line, X-chromosome deactivation was eroded, as is a common occurrence in iPSC lines (Anguera et al, 2012; Mekhoubad et al, 2012; Fukuda et al, 2021), so hits caused by differential expression of X-linked genes were disregarded. All the gene sets considered were either previously published (Mirigian et al, 2016; Phillips et al, 2018; Grandjean et al, 2019; Gorrell et al, 2022; Lamandé et al, 2023), or curated from previously described gene ontologies.

## GSEA

Differential expression results from DESeq2 were retrieved, and the "stat" column was used to pre-rank genes for GSEA analysis. These "stat" values reflect the Wald's test performed on read counts as modeled by DESeq2 using the negative binomial distribution. Genes that were not expressed were excluded from the analysis. GSEA (linux desktop version, v4.1) (Subramanian et al, 2005) was run in the pre-ranked mode against MSigDB 7.0 C5 (Gene Ontology) set, and ENSEMBL IDs were collapsed to gene symbols using the Human_ENSEMBL_Gene_ID_MSigDB.v7.4.chip (resulting in 25,162 and 26,199 genes for WT versus mutant sample in early and late time-point comparisons, respectively). In addition, a weighted scoring scheme, meandiv normalization, cutoffs on MSigDB signatures sizes (between 5 and 2,000 genes, resulting in 9,476 and 9,588 gene sets retained for early and late time-point WT versus mutant comparisons, respectively) were applied and 5,000 permutations were run for P-value estimation.

## Intracellular collagen-II quantification

A survey of confocal images of IHC samples stained for collagen-II was used to quantify the percent of cells with intracellularly retained collagen-II. 75 images of the different genotypes at different magnifications (40–100×) were arranged in a random order. Each genotype set consisted of at least four individual chondronoids. A total of at least 120 cells per genotype were assessed. Three independent and blinded participants were asked to count the total number of cells and the number of cells with intracellular staining in each image. The proportion of positive cells was averaged between all participants for each genotype, and the error bars represent the standard error of the mean. Statistical significance between genotypes was calculated using a heteroscedastic t test for each participant, and the largest of the three P-values obtained is displayed on the chart.

## Proteomics

### *Cell culture, transfections, and immunoprecipitation*

HT-1080 cells (ATCC) were cultured in complete DMEM supplemented with 10% FBS. Transfections of WT and Gly1170Ser *COL2A1*-encoding plasmids were performed using Transit-2020 transfection reagent (Mirus) according to manufacturer's instructions. The next day, media were replaced with fresh complete DMEM supplemented with 50 $\mu$M sodium ascorbate (Amresco). 2 d post-transfection, cells were harvested and washed three times with 1× PBS and cross-linked in the plate with 0.2 mM dithiobis(succinimidyl propionate) (DSP; Lomant's Reagent; Thermo Fisher Scientific) in PBS at RT for 30 min. Crosslinking was quenched by addition of 0.1 M Tris at pH 8.0 (final concentration) for 15 min at rt. Cells were scraped from plates in PBS using a cell scraper, then pelleted and resuspended in RIPA: 150 mM sodium chloride, 50 mM Tris–HCl, pH 7.5, 1% Triton X-100, 0.5% sodium deoxycholate, 0.1% SDS, protease inhibitor tablets and 1.5 mM phenylmethylsulfonyl fluoride (PMSF; Amresco). Cells were lysed for 30 min on ice, then centrifuged at 21,100$g$ for 15 min at 4°C. Supernatants were collected, quantified for protein content (BCA assay; Pierce), normalized, and incubated for 16 h with the HA antibody-conjugated beads, rotating end-over-end at 4°C. Samples were washed with RIPA 3× and eluted by boiling for 10 min in 300 mM Tris at pH 6.8 with 6% SDS and 600 mM DTT (Millipore) to release crosslinks. Eluates were then prepared for mass spectrometry analysis.

### *Mass spectrometry and interactome characterization*

Samples were reduced, alkylated, and digested with sequencing-grade trypsin (Promega) using an S-Trap Micro (Protifi) following standard protocol. After digestion, the samples were labeled with TMT 10 plex reagents (Thermo Fisher Scientific), according to the manufacturer's instructions. Briefly, the dried tryptic peptides were resuspended in 100 mM triethylammmonium bicarbonate, vortexed. Acetonitrile was added to each of the 10-plex TMT label reagents, vortexed, and briefly centrifuged. Peptides and labeling reagents were then combined in a 1:2 ratio and incubated for 1 h at RT, after which the reaction was quenched with 5% hydroxylamine for 15 min. Equal amounts of each channel were combined, dried by speed-vacuum, and resuspended in triethylammmonium bicarbonate. Samples were then fractionated with Pierce High pH Reversed-Phase Peptide Fractionation Kits (cat#84868) using standard protocols, then dried and resuspended in 0.2% formic acid for injection on LC–MS/MS. Labeled tryptic peptides were separated by reverse phase HPLC (Ultimate 3000; Thermo Fisher

Scientific) using a Thermo PepMap RSLC C$_{18}$ column (2 $\mu$m tip, 75 $\mu$m × 50 cm PN# ES903) over a gradient before nano-electrospray injection onto an Orbitrap Exploris 480 mass spectrometer (Thermo Fisher Scientific).

The mass spectrometer was operated in a data-dependent mode. The parameters for the full scan MS were: resolution of 120,000 across 375–1,600 $m/z$ and maximum IT of 25 ms. The full MS scan was followed by MS/MS for precursor ions in a 3 s cycle with a NCE of 32, dynamic exclusion of 30 s and resolution of 45,000. Raw mass spectral data files (.raw) were searched using Sequest HT in Proteome Discoverer (Thermo Fisher Scientific). Sequest search parameters were: 10 ppm mass tolerance for precursor ions; 0.05 D for fragment ion mass tolerance; two missed cleavages of trypsin; fixed modification were carbamidomethylation of cysteine and TMT modification on the lysines and peptide N-termini; variable modifications were methionine oxidation, methionine loss at the N-terminus of the protein, acetylation of the N-terminus of the protein, methylation of histidine, arginine, and glutamine, hydroxylation of proline, and methionine-loss plus acetylation of the protein N-terminus; total PTM per peptide was set to 8. The false discovery rate was set at 0.01. Data were searched against a human database and a common MS contaminant database made in house. Human Database: https://www.uniprot.org/proteomes/UP000005640.

Raw abundance values were used for quantification. First, abundance of each protein in the negative control was determined as the average abundance of the three no-transfection control conditions. High confidence interactors were identified as any protein detected with an abundance of at least 2× that of the negative control samples across all three replicates. Next, for each experimental sample, the abundance of each protein was normalized to the abundance of bait (collagen-II) to obtain a normalized abundance value. The fold enrichment was calculated as the ratio of the average of normalized abundance values in the Gly1170Ser conditions to the average of normalized abundance values in the WT conditions (Plate et al, 2019). Significance was evaluated using a homoscedastic $t$ test across the three biological replicates and the two-stage step-up method of Benjamini, Krieger, and Yekutieli with a desired FDR set to 10%.

### Supporting additional experimental procedures

#### *Gene editing of Gly1170Ser$^{+/+}$*

The nucleotide mutation c.3508 GGT > AGT (p.Gly1170Ser) was introduced in *COL2A1* in iPSCs derived from ATCC fibroblast line (ATCC CRL-1502) by CRISPR/Cas9, as previously described (Kung et al, 2020). Briefly, 1 × 10$^6$ WT iPSCs were electroporated (1100 V, 30 ms, 1 pulse) using the Neon transfection system and plated over four wells of a matrigel-coated six-well dish in E8 medium with 10 $\mu$M Y-27632. The medium was switched to E8 without Y-27632 the next day and changed every other day. Individual colonies were isolated and expanded in E8 medium. To identify potentially targeted clones, iPSC colonies were screened using allele-specific PCR primers COL2A1-1170scrF and COL2A1-1170mutR (Table S1). COL2A1-1170mutR should only bind the correctly targeted COL2A1-1170 mutation. Amplicons from the

control PCR using primers COL2A1-1170 scrF and COL2A1-1170 scrR that flank the introduced mutation (Table S1) were used to generate the antisense strand that was then Sanger-sequenced with the primer COL2A1-1170 seqR (Table S1) to confirm the mutation, clonality, and/or absence of indel mutations. Correctly targeted iPSC clones were amplified and confirmed free of mycoplasma contamination.

## Data Availability

All study data are included in the article or the SI/Appendix. The complete RNA-sequencing dataset is available on the GEO repository under GEO number GSE244375. Full proteomic data are available on MassIVE with accession number MSV000093517.

## Supplementary Information

## Acknowledgements

We extend our gratitude to Dr. Louise Kung, Lisa Sampurno, and Lynn Rowley at Murdoch Children's Research Institute for generously sharing their expertise and technical knowledge in iPSC culture, differentiation, and analysis. We thank Margaret Bisher, David Mankus, and Dr. Abigail Lytton-Jean at the Koch Institute for Integrative Cancer Research for technical support with transmission electron microscopy. We thank Brandyn Braswell at the Keck Microscopy Facility at the Whitehead Institute for technical support with confocal microscopy. We thank Noelani Kamelamela, Samuel Mildrum, and Dr. Stuart Levine at the MIT BioMicro Center for their assistance with RNA-sequencing. We thank Jimin Yoon, Jess Patrick, and Sorin Srinivasa for generously offering their time to quantify intracellular collagen. We also thank Andrea Yammine for contributions to graphic design. This work was supported by the National Institutes of Health (Grants 1R01AR071443 and R35GM136354 to MD Shoulders), a Research Grant from the G. Harold and Leila Y. Mathers Foundation (to MD Shoulders, SR Lamandé, and JF Bateman), the Australia National Health and Medical Research Council (GNT2003393 and GNT1146952 both to SR Lamandé and JF Bateman, and GNT1146902 to JF Bateman), and the Victorian Government's Operational Infrastructure Support Program. KM Yammine was supported by an NIH Ruth L. Kirschstein Predoctoral Fellowship (F31AR079263). S Kim was supported by a Kwanjeong Fellowship. This work was also supported in part by the National Cancer Institute (Koch Institute core) Grant P30-CA14051 and the National Institute of Environmental Health Sciences (core) Grant P30-ES002109. Work in the Novo Nordisk Foundation Center for Stem Cell Medicine is supported by a Novo Nordisk Foundation grant (NNF21CC0073729).

### Author Contributions

KM Yammine: conceptualization, data curation, formal analysis, validation, investigation, visualization, methodology, and writing—original draft, review, and editing.

S Mirda Abularach: formal analysis, validation, investigation, visualization, methodology, and writing—review and editing.

S Kim: data curation, formal analysis, validation, investigation, methodology, and writing—review and editing.

AA Bikovtseva: conceptualization, data curation, and writing—review and editing.

J Lilianty: data curation, formal analysis, investigation, methodology, and writing—review and editing.

VL Butty: software, formal analysis, visualization, methodology, and writing—review and editing.

RP Schiavoni: software, formal analysis, methodology, and writing—review and editing.

JF Bateman: conceptualization, data curation, formal analysis, supervision, funding acquisition, investigation, methodology, project administration, and writing—original draft, review, and editing.

SR Lamandé: conceptualization, data curation, formal analysis, funding acquisition, investigation, methodology, project administration, and writing—original draft, review, and editing.

MD Shoulders: conceptualization, data curation, formal analysis, supervision, funding acquisition, investigation, methodology, project administration, and writing—original draft, review, and editing.

## Conflict of Interest Statement

The authors declare that they have no conflict of interest.

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
