## [Reviewer comments · Life Science Alliance]

Life Science Alliance

ER procollagen storage defect without coupled unfolded protein response drives precocious arthritis

Kathryn Yamine, Sophia Mirda Abularach, Seo-yeon Kim, Agata Bikovtseva, Jinia Lilianty, Vincent Butty, Richard Schiavoni, John Bateman, Shireen Lamandé, and Matthew Shoulders

DOI: <https://doi.org/10.26508/lsa.202402842>

Corresponding author(s): Matthew Shoulders, Massachusetts Institute of Technology

Review Timeline:	Submission Date:	2024-05-28
	Editorial Decision:	2024-06-11
	Revision Received:	2024-06-21
	Accepted:	2024-06-26

Transaction Report:

Please note that the manuscript was reviewed at *Review Commons* and these reports were taken into account in the decision-making process at *Life Science Alliance*.

Reviews

Review #1

1. Evidence, reproducibility and clarity:

Evidence, reproducibility and clarity (Required)

The paper by Yammine et al addresses a major problem in peculiarities of genotype to phenotype manifestation in collagen II and chondrodysplasia. It is a lucid and comprehensive study detailing what they see as the fundamental mechanism of Gly1170 Ser mutated Col2a1 gene.

At the heart of the matter is the debunking of the results from a mouse model generated by liang et al (Plos one 2014) paper in which the authors suggested that the phenotype seen only homozygous mice (heterozygous mice appear normal), was related to ER stress -UPR-apoptosis cascade resulting in the chondrodysplasia. Yammine et al paper uses a different model, a robust human iPSC-based tissue, with a CRISPRed variant show that despite the ability of the variant chondrocytes to deposit a Gly1170Ser-substituted collagen II in both the hetero- and homozygous models, is not accompanied by any substantive UPR. The authors of this current paper also argue that their model system is most closely resemble the human context, where heterozygous individual show pathology.

I have sieved all the data related to this topic and have gone back to examine the data and what struck me was the repeated use of the phrase "slow to fold" in the current paper and wondered whether the element of "TIME" is as important in the chondrogenesis of either models and it is this element that generate the difference between the two results?

While iPSC-based tissue takes up to 44 days, a female mouse would have had two litters in this time and made many more growth plates. Could it be by "slowing" the chondrogenesis pathway, which is part of the procedure of differentiation of iPS cells into chondrocytes, the ER is not as "stressed" as in mouse development? I would like the authors to reflect and comment and put forward their view given UPR signaling pathways play a crucial role in chondrocytes in phases of high protein synthesis, e.g., during bone development by endochondral ossification (Journal of Bone Metabolism, vol. 24, no. 2, pp. 75-82, 2017).

This is not withstanding the good argument given by the authors in defending their robust results, namely that there is no evidence that the hydrophilic triple-helical domain of pro-collagen binds BiP, the main detector of accumulated misfolded proteins. What then do they make out of the immunostaining and qPCR with ER stress related genes in Liang et al paper?? I know that the data is not theirs but a comment on the indisputable data gives the reader a better understanding.

Although I understand the choice of cell lines for overexpression, the transfection of the HT-1080 cells using wild-type and Gly1170Ser COL2A1encoding plasmids are not a match to the in vivo model (variation of efficiency, etc.) the appearance of BiP even at a lower fold increase does not negate ER stress, as the authors acknowledge but more important is what other paracrine signals which triggers the UPR signally pathway which is not linked to BiP ? or an iPS system may lack? Is there anything else not only ATF6 α (activating transcription factor 6 alpha), but IRE1 α (inositol-requiring enzyme 1 alpha), and PERK (protein kinase RNA-like endoplasmic reticulum kinase). The authors have given us plenty of alternatives that are relevant, and they prepared us for yet another paper on articular cartilage using iPS tissue model which I am looking forward to.

2. Significance:

Significance (Required)

I think this paper is publishable and it is important in understanding the mechanism by which mutation in collagen type II affect chondrogenesis and therefore bone formation.

This paper will appeal to musculoskeletal scientist especially those who are interested in bone and its pathology.

It would be important for the authors to respond to the critique of "TIME" and speed of protein synthesis which create a duress in the ER pathway.

3. How much time do you estimate the authors will need to complete the suggested revisions:

Estimated time to Complete Revisions (Required)

(Decision Recommendation)

Less than 1 month

4. Review Commons values the work of reviewers and encourages them to get credit for their work. Select 'Yes' below to register your reviewing activity at Web of Science Reviewer Recognition Service (formerly Publons); note that the content of your review will not be visible on Web of Science.

Review #2

1. Evidence, reproducibility and clarity:

Evidence, reproducibility and clarity (Required)

System: The investigators have used a human iPSC chondrocyte model system to investigate the biochemistry of the Chondrodysplasia caused by the p.Gly1170Ser mutation in the type II collagen gene (COL2A1). They studied presumably homogeneous chondronoids formed by 3 cell lines they previously reported in which the chondrocytes were either homozygous wild type for the gene, homozygous for the Cas Crispr induced mutation or heterozygous for the two alleles (their refs 42-45). In addition, they utilized cultured HT1080 human fibrosarcoma cells transfected with wild type and mutant Col2A1 to study differences in the interactomes of the two proteins.

Analytic Parameters: They investigated the extracellular matrix formed by the three cells using collagen and proteoglycan staining and TEM and the transcriptional responses in chondronoids expressing the wild type and mutant genes.

Observations: matrix formation was defective in the two mutation bearing cell populations, reflecting defective fibril formation proportional to the abnormal gene dose. They found increased accumulations of post-translational modifications (hydroxylation, and O-glycosylation) on the mutant collagen extracted from the chondronoids and EM evidence of collagen retention in the ER. They studied the comparative transcriptional profiles in the three phenotypes and failed to find a profound UPR response late in culture and only a mild upregulation of UPR genes in the young cultures. They could not find evidence for activation of the ISR except in the homozygous mutant cells. Using transfected HT-1080 cells (previously shown by these investigators not to express endogenous pro-collagen II but able to synthesize transfected pro-collagen genes) they were able to study the comparative wt and mutant pro-collagen interactomes.

Conclusions: They conclude that the p.gly1170ser mutation in Col2A1 results in abnormal folding which results in trapping of the protein in the ER and some interaction with cellular elements of the proteostatic response. They concluded that the cellular proteostasis machinery can recognize slow-folding Gly1170Ser through increased interactions with certain ER network components but not in the same fashion that has been described for liver cells producing mutated versions of high volume secreted proteins.

****Major comments:****

Their first conclusion, stated in the abstract, "Biochemical characterization reveals that Gly1170Ser procollagen-II is notably slow to fold and secrete." that the mutant polypeptide chain is slower folding than the wild type chain is based on the premise that the longer the chains are in the ER the greater the degree of lysine hydroxylation and O-glycosylation. Although this may be true, they do not provide a reference and I could not find a definitive description of the phenomenon. Their reference 48 only discusses the occurrence of intracellular post-translational modification of the lysines and continuing modification extracellularly but does not relate these phenomena to the rate at which the peptides traverse the cell. I think the reader would benefit from seeing experiments in which the rate of folding and secretion of the wild type and mutant chains are measured and the degree of post-translational modification are compared. Cabral WA et al showed differences in collagen folding and secretion rates in cyclophilin wt, knockouts and heterozygotes osteoblasts and fibroblasts by western blots. (2014) Abnormal Type I Collagen Post-translational Modification and Crosslinking in a Cyclophilin B KO Mouse Model of Recessive Osteogenesis Imperfecta. PLoS Genet 10(6): e1004465. doi:10.1371 / journal.pgen. 1004465). Performing such experiments in their chondronoids would confirm the authors' interpretation that the increased post-translational modification portrayed in their figure 4 reflects slowed folding and secretion related to the mutation.

I think Figure 4 needs more explanation for the reader. While, as expected, the homozygous mutant band is much slower than the homozygous wild type band, in the heterozygotes the band is intermediate rather than showing a discrete mixture of wild type and mutant proteins, reflecting different degrees of post-translational modification. Is this a function of mixed triple helices with heterogeneous degrees of post-translational modification? It deserves more comment, since the argument relating the degree of post-translational modification to the rate of folding is dependent on this observation. It would also be helpful to show the whole gel with collagen II markers.

Another approach to the question of intracellular accumulation due to a slow rate of folding of the mutant collagen would be to perform pulse chase labeling of the three types of chondronoids with radiolabeled amino acids and sugars and processing the media and lysates with analysis using antibodies specific for the two collagen chain types. Given the authors extensive experience in studying collagen biosynthesis (e.g. Chan et al J. Biochem. Biophys. Methods 36 (1997) 11-29), such a supporting study would firmly establish whether the rate of folding/secretion differs between the wt and the homozygous and heterozygous chondroidinomas. Until the slow folding can be directly demonstrated in a quantitative fashion rather than by monitoring the secondary phenomenon of post-translational modification the hypothesis remains unproven.

Another issue that does not appear to be addressed is the consequence of having misfolded collagen chains in the dilated ER. Liang et al, using mice transgenic for one or two copies of the mutant human gene showed apoptosis in the homozygotes but not in the hets a finding similar to that of Kimura et al using transgenics carrying a different human COL2A1 mutation. Okada et al, using chondrocytes converted from human fibroblasts with clinical collagenopathy (heterozygous), although not the same mutation as in the present study, showed dilated ER and some

level of apoptosis in the cultured cells. Hintze et al, examining chondrocytes expressing different mutants associated with different forms of spondyloepiphyseal dysplasia, suggested that the degree of stability of the mutations might determine whether apoptosis occurred, i.e. the thermolabile p.R989C was associated with apoptosis while cells expressing the more thermostable mutants p.275C, P.719C and p.G853E did not reveal any evidence for ongoing apoptosis R989. Is it possible that the smaller size of the homozygous chondronoids reflect fewer cells rather than less matrix (or both) as result of apoptosis? Examination of the chondronoids with reagents for caspase 3 or TUNEL staining. One could also measure by Col/DNA ratio in wt, hets and homos. It might also have been useful for these experiments been more quantitative, i.e. by cell sorting rather than by eye. Would ImageJ software been helpful? It is also unclear as to the conformation of chains trapped in the ER. There are many examples in which the natural tendency of misfolded proteins is to aggregate. This is certainly true in the neurodegenerative diseases. While at the magnification used here in the TEM's the ER inclusions appear homogeneous and amorphous, perhaps at higher magnification/resolution a more discrete structure might be seen.

While the choice of time points for the transcriptional analysis, i.e. early and late seems well thought out, the lack of a significant response may be due to the timing and it might have been useful to do earlier or later time points or intermediate time points in case the response was transient, particularly since other laboratories have reported UPR activation and abnormalities in the context of the silencing of Xbp1, the spliced form of which is a major driver of at least one arm of the UPR. The notion that pro-collagen is largely hydrophilic without the potential for exposure of hydrophobic regions that might engage BiP, thus is not sensitive to BiP sensing, is interesting. Is it possible that the tendency of the mutant polypeptides to form the triple helix which in itself acts as kind of a self chaperoning structure? Looking at the kinetics of assembly inside the cell, see suggestions above, might provide further insight into the process beyond that obtained by looking at the modified state of the lysines.

****Minor comments:****

As I mentioned above, while the transcriptional interactome experiments are computationally sophisticated the cell biology and biochemistry would benefit from more and better quantitation.

The paper is written in a style in which results and discussion are intermingled. Personally I prefer that the introductions are short, the results clearly and briefly presented and the discussion deals with the interpretation and conclusions. I thought that whole paragraphs could have been omitted.
e.g. in the introduction

Omit paragraph

"The fibrillar.....achondrogenesis type II"

Omit paragraph

"Conventional and... for example."

Omit

"Excitingly.....in vitro and in vivo (36)."

Results:

First paragraph repeats last paragraph of introduction and not necessary in one place or the other, condense.
Figure 2 by eye MGP (Matrix gla protein inhibits vascular calcification of type II collagen) seems highly over-expressed in the homozygous mutants; MGP is supposedly an inhibitor of calcification, does its over-expression here reflect something about the adequacy of the matrix

Figure 5 is good but can it be confirmed by quantitative biochemistry?

Did you stain with antibodies to other ER resident chaperones other than calreticulin?

Do cells with large amounts of intracellular G1170S die?

Does higher magnification EM reveal any structure of the material within the dilated ER?

Are there any inflammatory cells in the Chondronoids? To respond to aberrant proteins?

Paragraph

"Bypassing the UPR.....often do not" Is discussion not results

****Referees cross-commenting****

Are other reviewers concerned about the precise definition of slowed folding rather than utilizing the degree of post-translational modification as a surrogate?

2. Significance:

Significance (Required)

The experimental system described here is clearly the wave of the present. Generating human ipSC's of different lineages is now being exploited to study a variety of disorders, to achieve better understanding of pathogenesis at the molecular level to serve as appropriate models for drug development, particularly in the context of high throughput screening. In addition, as in this case, relatively rare autosomal dominant disorders with phenotypes that resemble more common sporadic disease, may allow the development of treatments that are relevant for the sporadic disorder. While it is likely that the osteoarthritis that develops in the carriers of the COL2A1 mutations is a function of the host response to the aberrant mechanics resulting from the defective extra-cellular matrix caused by the mutation, having a pure system in which the primary defect can be corrected and the predisposing matrix deficit reversed, could allow normal reparative processes to mitigate the functional joint disability. While the transgenic mice are useful as a disease model, they represent not only the expression of the primary defect but the host pathophysiologic response to that defect, i.e. in this case how the mouse responds to the defective matrix state and whether those responses add additional pathogenic factors to the disease course. Having a tool in which to relatively assess the pure chondrocyte effect should allow more granular analysis of the primary process.

Their findings reinforce the notion that involvement of the UPR as well as the other arms of the proteostatic response in chondrocytes expressing a variety of mutant collagens suggests a degree of heterogeneity, perhaps depending on the mutation involved. While I do not believe that their current data prove or rigorously test their proposed hypothesis, i.e. that "perhaps due to the pathologic substitution occurring within a triple-helical domain that lacks hydrophobic character, this ER protein accumulation is not recognized by cellular stress responses, such as the unfolded protein response", it is worth considering.

Given the fact that this is a relatively small field with a variety of observations concerning the role of proteostasis and the UPR in particular which seem to vary depending on the system, i.e. transgenic mice, transfected fibroblasts, the chondroidomas, these observations particularly with additional biochemistry to confirm their notions regarding folding rates etc, represent a useful technical addition to the field and should be interesting for people working on collagen biology, arthritis and protein folding.

I am not a collagen biologist hence my knowledge of some of the nuances of collagen biology may not be extensive. My own areas of interest include the assembly of multi-peptide proteins (such as immunoglobulins) for secretion; the mechanisms that allow them to exit the cell and the aggregation of misfolded proteins as exemplified by the amyloidoses and other forms of clinically relevant protein aggregation. Hence, I am very familiar with tissue culture, transgenic animals as disease models, studies of protein aggregation, and as a former rheumatologist, osteoarthritis.

3. How much time do you estimate the authors will need to complete the suggested revisions:

Estimated time to Complete Revisions (Required)

(Decision Recommendation)

Between 1 and 3 months

4. Review Commons values the work of reviewers and encourages them to get credit for their work. Select 'Yes' below to register your reviewing activity at Web of Science Reviewer Recognition Service (formerly Publons); note that the content of your review will not be visible on Web of Science.

Authors' Response to Reviewers

1. General Statements [optional]

This section is optional. Insert here any general statements you wish to make about the goal of the study or about the reviews.

Reviewer #1

The paper by Yammine et al addresses a major problem in peculiarities of genotype to phenotype manifestation in collagen II and chondrodysplasia. It is a lucid and comprehensive study detailing what they see as the fundamental mechanism of Gly1170 Ser mutated Col2a1 gene.

At the heart of the matter is the debunking of the results from a mouse model generated by Liang et al (Plos one 2014) paper in which the authors suggested that the phenotype seen only homozygous mice (heterozygous mice appear normal), was related to ER stress -UPR-apoptosis cascade resulting in the chondrodysplasia. Yammine et al paper uses a different model, a robust human iPSC-based tissue, with a CRISPRed variant show that despite the ability of the variant chondrocytes to deposit a Gly1170Ser-substituted collagen II in both the hetero- and homozygous models, is not accompanied by any substantive UPR. The authors of this current paper also argue that their model system is most closely resemble the human context, where heterozygous individual show pathology.

We appreciate the Reviewer highlighting the significance of this manuscript addressing a major issue in the field.

I have sieved all the data related to this topic and have gone back to examine the data and what struck me was the repeated use of the phrase "slow to fold" in the current paper and wondered whether the element of "TIME" is as important in the chondrogenesis of either models and it is this element that generate the difference between the two results? While iPSC-based tissue takes up to 44 days, a female mouse would have had two litters in this time and made many more growth plates. Could it be by "slowing" the chondrogenesis pathway, which is part of the procedure of differentiation of iPSC cells into chondrocytes, the ER is not as "stressed" as in mouse development? I would like the authors to reflect and comment and put forward their view given UPR signaling pathways play a crucial role in chondrocytes in phases of high protein synthesis, e.g., during bone development by endochondral ossification (Journal of Bone Metabolism, vol. 24, no. 2, pp. 75-82, 2017).

The Reviewer here emphasizes a likely benefit of the human model that we had not previously considered, as differentiation and growth in our model are indeed far more similar to humans than the rapid timeline in mice. We also note that the evidence that collagen is slow to fold comes from the gold standard assay in the field for collagen folding rate, a point discussed in greater detail in response to Reviewer 2's query (see below).

The literature evidence does indicate that transient UPR signaling is relevant for chondrogenesis. We selected UPR timepoints that do not interface with the differentiation process, but rather the tissue deposition process while chondrocytes are still actively depositing and maintaining the extracellular matrix, to be able to distinguish a physiological transient UPR during differentiation from a potential chronic and possibly pathologic one. We now clarify this point in the manuscript (see text below).

“These timepoints were selected to reflect an early and a late stage of cartilage maturation, but with both timepoints harvested post-chondrogenesis so as not to interfere with the physiologic transient UPR activation that can be important in that process.”

This is not withstanding the good argument given by the authors in defending their robust results, namely that there is no evidence that the hydrophilic triple-helical domain of pro-collagen binds BiP, the main detector of accumulated misfolded proteins. What then do they make out of the immunostaining and qPCR with ER stress related genes in Liang et al paper?? I know that the data is not theirs but a comment on the indisputable data gives the reader a better understanding.

It is critical to note that the evidence for ER stress that induces the UPR is, at best, exceptionally weak for heterozygotes in the Liang *et al* paper, and arguably also weak for homozygotes. Liang *et al* observed, via quantitative PCR, that the mRNA levels of just Chop (which is also a marker of the integrated stress response and not a good readout for UPR activity) and ATF6 (whose RNA-level upregulation is not a standard marker of the UPR) were significantly upregulated in

the disease-relevant heterozygous mice – given that other (more valid) UPR markers were not altered, this is not so different from our observation of a lack of UPR in the disease-relevant **heterozygotes**.

A somewhat more comprehensive set of UPR markers, including Chop, Xbp1(Total and Spliced), Grp78 (BiP), ATF4, and ATF6, was significantly upregulated only in homozygous mice compared to wild-type. PERK is one of many kinases upstream of ATF4 and Chop that can be activated by a variety of processes (the pathway is part of the integrated stress response, for example). Moreover, transcriptional upregulation of ATF4 (which is actually induced translationally, not transcriptionally) and ATF6 (which is actually induced proteolytically, not transcriptionally) are not normally used to read out UPR activation, so it is not so clear to us that a robust UPR was induced even in homozygotes. Moreover, there was not a substantial increase in Xbp1-S (S = spliced) **relative** to Xbp1-T (T= total) in the study of homozygous mice, which is the most appropriate measure of UPR activation – rather than change in Xbp1-T and Xbp1-S. The use of mostly non-standard genes to assess UPR induction, the weak upregulation of BiP (2.5-fold), and the unchanged ratio of Xbp1-S to Xbp1-T raise some questions regarding UPR induction even in the homozygotes. Regardless of these homozygote data, as noted above, the evidence for a UPR in heterozygotes is very weak, despite ER stress being the focus of the Liang *et al* paper.

With respect to immunostaining, Liang *et al* observed that tissue from homozygous mice (but not heterozygotes) contained significantly more apoptotic cells. Apoptosis could be a result of chronic, unresolved UPR signaling, but it could also result from any number of other pathways and is certainly not direct evidence for UPR-inducing ER stress. Additionally, for the homozygote apoptosis assay, Liang *et al* do not note how many mice were analyzed for each genotype, a value they did report for their other assays. While examining multiple sections for each genotype is valuable (they state ≥ 10), the assessment of biological replicates (additional mice) seems critical to confidently reach a conclusion.

Although I understand the choice of cell lines for overexpression, the transfection of the HT-1080 cells using wild-type and Gly1170Ser COL2A1 encoding plasmids are not a match to the in vivo model (variation of efficiency, etc.) the appearance of BiP even at a lower fold increase does not negate ER stress, as the authors acknowledge but more important is what other paracrine signals which triggers the UPR signaling pathway which is not linked to BiP? or an iPS system may lack? Is there anything else not only ATF6 α (activating transcription factor 6 alpha), but IRE1 α (inositol-requiring enzyme 1 alpha), and PERK (protein kinase RNA-like endoplasmic reticulum kinase).

Our finding that the UPR is not activated is based on comprehensive RNA-sequencing performed in the physiologically more relevant iPSC-derived chondrocyte, as opposed to the tumor cell line HT-1080. Our interactomic finding that BiP interacts to the same extent with wild-type and Gly1170Ser procollagen-II (in HT-1080 cells) strongly supports our proposal that the reason the UPR is not activated is that BiP fails to recognize unfolded triple-helical domains.

We note that, although HT-1080 cells are not a perfect match, they are the most accessible option for interactome-based studies. Because there is no MS-grade antibody for collagen-II IP, we need to IP a transfected, tagged collagen. We cannot do this in chondronoids, or in isolated chondrocytes that transfect poorly and rapidly dedifferentiate. Critically, Prockop and co-workers extensively validated HT-1080 cells as a platform for fibrillar collagen biochemical studies in *Matrix* **1993**, 13, 399. Our own lab further characterized their capacity to properly handle fibrillar collagen variants in great molecular detail in *ACS Chem Biol* **2016**, 11, 1408.

Since our chondronoid system contains only chondrocyte cells, as is the case in cartilage, the cells can receive paracrine signals from other chondrocytes, but not other cell types. In joints within a whole animal, it is true that paracrine crosstalk occurs between different cell types of different tissues, including inflammatory cells for example. The chondronoid is very useful for elucidating the defects that occur at the chondrocyte-level, without confounding secondary effects. At the chondrocyte-level, Gly1170Ser-substituted procollagen-II does not activate the UPR.

The Reviewer's comment regarding the absence of paracrine signals in an iPSC-based system is well-taken, and we added discussion as follows:

“These observations indicate that the chondrocytes were not raising such stress responses, at least when examined in the absence of paracrine signals from other cell types in the joint.”

The authors have given us plenty of alternatives that are relevant, and they prepared us for yet another paper on articular cartilage using iPSC tissue model which I am looking forward to.

We are also excited about the upcoming potential of this model system!

Significance

I think this paper is publishable and it is important in understanding the mechanism by which mutation in collagen type II affect chondrogenesis and therefore bone formation. This paper will appeal to musculoskeletal scientist especially those who are interested in bone and its pathology. It would be important for the authors to respond to the critique of "TIME" and speed of protein synthesis which create a duress in the ER pathway.

We greatly appreciate the Reviewer's comment again on the significance of this work, and their scholarly input which has substantially improved the paper. We hope they will agree that the manuscript is now ready for publication.

Reviewer #2

Evidence, reproducibility and clarity

System: The investigators have used a human iPSC chondrocyte model system to investigate the biochemistry of the Chondrodysplasia caused by the p.Gly1170Ser mutation in the type II collagen gene (COL2A1). They studied presumably homogeneous chondronoids formed by 3 cell lines they previously reported in which the chondrocytes were either homozygous wild type for the gene, homozygous for the Cas Crispr induced mutation or heterozygous for the two alleles (their refs 42-45). In addition, they utilized cultured HT1080 human fibrosarcoma cells transfected with wild type and mutant Col2A1 to study differences in the interactomes of the two proteins.

Analytic Parameters: They investigated the extracellular matrix formed by the three cells using collagen and proteoglycan staining and TEM and the transcriptional responses in chondronoids expressing the wild type and mutant genes.

Observations: matrix formation was defective in the two mutation bearing cell populations, reflecting defective fibril formation proportional to the abnormal gene dose. They found increased accumulations of post-translational modifications (hydroxylation, and O-glycosylation) on the mutant collagen extracted from the chondronoids and EM evidence of collagen retention in the ER. They studied the comparative transcriptional profiles in the three phenotypes and failed to find a profound UPR response late in culture and only a mild upregulation of UPR genes in the young cultures. They could not find evidence for activation of the ISR except in the homozygous mutant cells.

Using transfected HT-1080 cells (previously shown by these investigators not to express endogenous pro-collagen II but able to synthesize transfected pro-collagen genes) they were able to study the comparative wt and mutant pro-collagen interactomes.

Conclusions: They conclude that the p.gly1170ser mutation in Col2A1 results in abnormal folding which results in trapping of the protein in the ER and some interaction with cellular elements of the proteostatic response. They concluded that the cellular proteostasis machinery can recognize slow-folding Gly1170Ser through increased interactions with certain ER network components but not in the same fashion that has been described for liver cells producing mutated versions of high volume secreted proteins.

We appreciate this careful summary of our work.

Major comments:

Their first conclusion, stated in the abstract, "Biochemical characterization reveals that Gly1170Ser pro-collagen-II is notably slow to fold and secrete." that the mutant polypeptide chain is slower folding than the wild type chain is based on the premise that the longer the chains are in the ER the greater the degree of lysine hydroxylation and O-glycosylation. Although this may be true, they do not provide a reference and I could not find a definitive description of the phenomenon. Their reference 48 only discusses the occurrence of intracellular post-translational modification of the lysines and continuing modification extracellularly but does not relate these phenomena to the rate at which the peptides traverse the cell. I think the reader would benefit from seeing experiments in which the rate of folding and secretion of the wild type and mutant chains are measured and the degree of post-translational modification are compared. Cabral WA et al showed differences in collagen folding and secretion rates in cyclophilin wt, knockouts and heterozygotes osteoblasts and fibroblasts by western blots. (2014) Abnormal Type I Collagen Post-translational Modification and Crosslinking in a Cyclophilin B KO Mouse Model of Recessive Osteogenesis Imperfecta. PLoS Genet 10(6): e1004465. doi:10.1371 / journal.pgen. 1004465). Performing such experiments in their chondronoids would confirm the authors' interpretation that the increased post-translational modification portrayed in their figure 4 reflects slowed folding and secretion related to the mutation.

We apologize for failing to provide essential background references and information to assess our assay for slow folding/secretion of procollagen. In fact, slow migration on SDS-PAGE is not only a widely used assay for comparing the rate of folding of procollagens, it has also remained the gold standard in the field for the past forty years. The studies cited below are some of the seminal papers in the field linking collagen's rate of folding with its extent of posttranslational modifications and its electrophoretic mobility. We have now updated our citations accordingly.

1. Bateman, J.F.; Mascara, T.; Chan, D.; Cole, W.G. "Abnormal type I collagen metabolism by cultured fibroblasts in lethal perinatal osteogenesis imperfecta" *Biochem J* **1984**, *217*, 103.
2. Bonadio, J.; Holbrook, K.A.; Gelinias, R.E.; Jacob, J.; Byers, P.H. "Altered triple helical structure of type I procollagen in lethal perinatal osteogenesis imperfecta" *J Biol Chem* **1985**, *260*, 1734.
3. Bateman, J.F.; Chan, D.; Mascara, T.; Rogers, J.G.; Cole, W.G. "Collagen defects in lethal perinatal osteogenesis imperfecta" *Biochem J* **1986**, *240*, 699.
4. Godfrey, M.; Hollister, D.W. "Type II achondrogenesis-hypochondrogenesis: Identification of abnormal type II collagen" *Am J Hum Genet* **1988**, *43*, 904.

The basis for this collagen-specific assay of folding rate is that the ER-localized procollagen proline and lysine hydroxylases require monomeric collagen strands as substrates, and cannot accommodate a folded triple helix in their active sites. Thus, accumulation of post-translational modifications on collagen depends on the procollagen triple-helical domain's residence time as an unfolded monomeric region of the assembling triple-helical trimer within the ER. Some fraction of the hydroxylated lysines are later glycosylated, which slows migration on SDS-PAGE gels. We have now clarified our slow folding conclusion with more precise references and discussion in the manuscript.

Pulse-chase experiments like those suggested by the Reviewer would indeed be beneficial if they were possible in this system, but they simply are not. Although it might be possible to soak in a radiolabeled amino acid over a short time period, the assay still relies on separating the cell fraction from the secreted fraction. This is possible in monolayer cultures, but in a chondronoid composed of complex cartilage and cells we have no way to do it. One could propose that we extract the chondrocytes and then do the pulse-chase in a monolayer culture, but this unfortunately is also not possible as chondrocytes do not behave well outside the tissue setting and rapidly differentiate into other cell types. Fortunately, the procollagen overmodification assay is a widely used and well-accepted measure of slow folding, and thus addresses the issue.

I think Figure 4 needs more explanation for the reader. While, as expected, the homozygous mutant band is much slower than the homozygous wild type band, in the heterozygotes the band is intermediate rather than showing a discrete mixture of wild type and mutant proteins, reflecting different degrees of post-translational modification. Is this a function of mixed triple helices with heterogeneous degrees of post-translational modification? It deserves more comment, since the argument relating the degree of post-translational modification to the rate of folding is dependent on this observation. It would also be helpful to show the whole gel with collagen II markers.

We modified **Figure 4** to show the whole gel (in the SI, see **Fig. S3**) and molecular weight markers. It also shows the wild-type collagen-II band. Most of the procollagen produced by the heterozygote is heterotrimeric for the disease-causing substitution (>87% of trimers will contain at least one mutant chain and thus experience delayed folding) and, therefore, the diffuse banding structure is to be expected. Further, we would speculate that in these challenged ER, even the folding of wild-type only trimers is impaired. The Reviewer's comment suggests there may be some basis for that speculation. We added a note to this effect.

"The presence of a single broad, slow-migrating band as opposed to distinctive overmodified mutant versus normally modified wild-type strands is due to fact that the vast majority of trimers formed in heterozygotes (>85%) contain at least one Gly1170Ser strand that delays triple-helix folding."

Another approach to the question of intracellular accumulation due to a slow rate of folding of the mutant collagen would be to perform pulse chase labeling of the three types of chondronoids with radiolabeled amino acids and sugars and processing the media and lysates with analysis using antibodies specific for the two collagen chain types. Given the authors extensive experience in studying collagen biosynthesis (e.g. Chan et al *J. Biochem. Biophys. Methods* **36 (1997) 11-29), such a supporting study would firmly establish whether the rate of folding/secretion differs between the wt and the homozygous and heterozygous chondroidinomas. Until the slow folding can be directly demonstrated in a quantitative fashion rather than by monitoring the secondary phenomenon of post-translational modification the hypothesis remains unproven.**

Discussed above in response to the Reviewer's earlier suggestion of pulse-chase and question regarding the post-translational modification assay, unfortunately the pulse-chase experiment is infeasible. Fortunately, the modification-based assay is already the gold standard in the collagen field.

Another issue that does not appear to be addressed is the consequence of having misfolded collagen chains in the dilated ER. Liang et al, using mice transgenic for one or two copies of the mutant human gene showed apoptosis in the homozygotes but not in the hets a finding similar to that of Kimura et al using transgenics carrying a different human COL2A1 mutation. Okada et al, using chondrocytes converted from human fibroblasts with clinical collagenopathy (heterozygous), although not the same mutation as in the present study, showed dilated ER and some level of apoptosis in the cultured cells. Hintze et al, examining chondrocytes expressing different mutants associated with different forms of spondyloepiphyseal dysplasia, suggested that the degree of stability of the mutations might determine whether apoptosis occurred, i.e. the thermolabile p.R989C was associated with apoptosis while cells expressing the more thermostable mutants p.275C, P.719C and p.G853E did not reveal any evidence for ongoing apoptosis R989. Is it possible that the smaller size of the homozygous chondronoids reflect fewer cells rather than less matrix (or both) as result of apoptosis? Examination of the chondronoids with reagents for caspase 3 or TUNEL staining. One could also measure by Col/DNA ratio in wt, hets and homos. It might also have been useful for these experiments been more quantitative, i.e. by cell sorting rather than by eye. Would ImageJ software been helpful?

We greatly appreciate this suggestion. We now added results of TUNEL assays performed on sections of the chondronoids (see **Fig. 8**), including quantification of the results. Notably, we do not observe a significant difference in apoptosis between genotypes at the timepoint considered. This result is also supported by our transcriptional data, where we do not observe upregulation of apoptosis-related pathways, via the UPR or otherwise.

It is also unclear as to the conformation of chains trapped in the ER. There are many examples in which the natural tendency of misfolded proteins is to aggregate. This is certainly true in the neurodegenerative diseases. While at the magnification used here in the TEM's the ER inclusions appear homogeneous and amorphous, perhaps at higher magnification/resolution a more discrete structure might be seen.

From collagen-II immunohistochemistry confocal images, the intracellular collagen appears sometimes as aggregated puncta, and in other cases more diffuse and amorphous. Given this heterogeneity, we were not able to readily obtain clear additional structural characterization of the intracellular procollagen-II fraction.

While the choice of time points for the transcriptional analysis, i.e. early and late seems well thought out, the lack of a significant response may be due to the timing and it might have been useful to do earlier or later time points or intermediate time points in case the response was transient, particularly since other laboratories have reported UPR activation and abnormalities in the context of the silencing of Xbp1, the spliced form of which is a major driver of at least one arm of the UPR.

While our RNA-sequencing results at the specific timepoints we chose cannot rule out a transient activation of the UPR, they do indicate that chronic, unresolved UPR signaling is not the underlying cause of pathology, which is the main point we are making.

The notion that pro-collagen is largely hydrophilic without the potential for exposure of hydrophobic regions that might engage BiP, thus is not sensitive to BiP sensing, is interesting. Is it possible that the tendency of the mutant polypeptides to form the triple helix which in itself acts as kind of a self chaperoning structure? Looking at the kinetics of assembly inside the cell, see suggestions above, might provide further insight into the process beyond that obtained by looking at the modified state of the lysines.

We believe this notion is very strongly supported by the interactomic experiment showing that BiP fails to preferentially engage the poorly folding triple-helical variant. There are, however, many other chaperones and folding enzymes that assist collagen folding, including prolyl isomerases and Hsp47. Hence, it is not clear to us that substantial self-chaperoning occurs. Still, the self-chaperoning idea is intriguing, and we will note that prior work does indicate that triple-helical domains of individual procollagen polypeptides are strongly pre-organized for triple-helix formation (for a review, see *Annu Rev Biochem* **2009**, 78, 929). That said, we hesitate to speculate here on the self-chaperoning idea without additional evidence.

Minor comments:

As I mentioned above, while the transcriptional interactome experiments are computationally sophisticated the cell biology and biochemistry would benefit from more and better quantitation.

We have included quantitation of the extent of intracellular procollagen accumulation and the extent of apoptotic cells, which we hope helps to address this point.

The paper is written in a style in which results and discussion are intermingled. Personally I prefer that the introductions are short, the results clearly and briefly presented and the discussion deals with the interpretation and conclusions. I thought that whole paragraphs could have been omitted.

e.g. in the introduction

Omit paragraph

"The fibrillar.....achondrogenesis type II"

Omit paragraph

"Conventional and... for example."

Omit

"Excitingly.....in vitro and in vivo (36)."

Results:

First paragraph repeats last paragraph of introduction and not necessary in one place or the other, condense.

We appreciate this feedback and have accordingly edited the manuscript for clarity and brevity, which includes deleting or significantly shortening all the paragraphs indicated by the Reviewer. These improvements are indicated in the track-changes version of the manuscript we resubmitted.

Figure 2 by eye MGP (Matrix gla protein inhibits vascular calcification of type II collagen) seems highly over-expressed in the homozygous mutants; MGP is supposedly an inhibitor of calcification, does its over-expression here reflect something about the adequacy of the matrix

Overexpression of MGP could indeed reflect a defect in the matrix of the homozygous variants. It is also likely a reflection of the delayed hypertrophy and maturation observed in the homozygous variants, as matrix calcification is a step in the endochondral ossification process. We did not follow-up on this particular observation, as it is exclusively observed in the less clinically relevant homozygous variant. We added a note to the manuscript to capture the Reviewer's point about MGP, as below:

"The upregulation in the homozygous system of Matrix Gla Protein (MGP) (Fig. 2A), which inhibits vascular calcification of the matrix *in vivo*, further supports the delay in hypertrophy, and could lead to differences in the biomechanical properties of the matrix."

Figure 5 is good but can it be confirmed by quantitative biochemistry?

We have included quantitation of the extent of intracellular procollagen accumulation and the extent of apoptotic cells.

Did you stain with antibodies to other ER resident chaperones other than calreticulin?

Yes, we also stained the ER with PDI. However, the chondronoids require extensive optimization for immunostaining and we could obtain much better images using the ER marker for calreticulin, hence our choice of images to present in the manuscript.

Do cells with large amounts of intracellular G1170S die?

As indicated by the newly included TUNEL data, interestingly, even cells expressing exclusively the Gly1170Ser variant of procollagen-II do not seem to apoptose at a significantly higher rate than wild-type, at least at the timepoint considered. As mentioned above, we added these data as Fig. 8, and added discussion of these results and methodology in the relevant sections of the manuscript.

Does higher magnification EM reveal any structure of the material within the dilated ER?

We have so far not been able to use EM to obtain higher-resolution insight into intracellular procollagen structures, but we will work on this idea in future studies.

Are there any inflammatory cells in the Chondronoids? To respond to aberrant proteins?

There should not be any such cells present in the chondronoids, and we indeed do not observe any inflammatory response. As noted in the response to Reviewer 1, we added discussion regarding the absence of paracrine signals in this type of model system, which we do believe has major advantages for biochemical studies like those performed here.

Paragraph

"Bypassing the UPR.....often do not" Is discussion not results

Corrected, thanks.

Significance

The experimental system described here is clearly the wave of the present. Generating human ipSC's of different lineages is now being exploited to study a variety of disorders, to achieve better understanding of pathogenesis at the molecular level to serve as appropriate models for drug development, particularly in the context of high throughput screening. In addition, as in this case, relatively rare autosomal dominant disorders with phenotypes that resemble more common sporadic disease, may allow the development of treatments that are relevant for the sporadic disorder. While it is likely that the osteoarthritis that develops in the carriers of the COL2A1 mutations is a function of the host response to the aberrant mechanics resulting from the defective extra-cellular matrix caused by the mutation, having a pure system in which the primary defect can be corrected and the predisposing matrix deficit reversed, could allow normal reparative processes to mitigate the functional joint disability. While the transgenic mice are useful as a disease model, they represent not only the expression of the primary defect but the host pathophysiologic response to that defect, i.e. in this case how the mouse responds to the defective matrix state and whether those responses add additional pathogenic factors to the disease course. Having a tool in which to relatively assess the pure chondrocyte effect should allow more granular analysis of the primary process.

We appreciate the Reviewer's careful and enthusiastic assessment of the significance of our work.

Their findings reinforce the notion that involvement of the UPR as well as the other arms of the proteostatic response in chondrocytes expressing a variety of mutant collagens suggests a degree of heterogeneity, perhaps depending on the mutation involved. While I do not believe that their current data prove or rigorously test their proposed hypothesis, i.e. that "perhaps due to the pathologic substitution occurring within a triple-helical domain that lacks hydrophobic character, this ER protein accumulation is not recognized by cellular stress responses, such as the unfolded protein response", it is worth considering.

We provide that hypothesis as a reasonable explanation for the absence of a UPR, and it is strongly supported by our interactomic studies. Furthermore, neither we nor others have found evidence for BiP binding the triple helical domain of procollagen in any other studies. Still, that hypothesis is not the core point of the paper and we do appreciate the Reviewer's perspective.

Given the fact that this is a relatively small field with a variety of observations concerning the role of proteostasis and the UPR in particular which seem to vary depending on the system, i.e. transgenic mice, transduced fibroblasts, the chondroidomas, these observations particularly with additional biochemistry to confirm their notions regarding folding rates etc, represent a useful technical addition to the field and should be interesting for people working on collagen biology, arthritis and protein folding.

I am not a collagen biologist hence my knowledge of some of the nuances of collagen biology may not be extensive. My own areas of interest include the assembly of multi-peptide proteins (such as immunoglobulins) for secretion; the mechanisms that allow them to exit the cell and the aggregation of misfolded proteins as exemplified by the amyloidoses and other forms of clinically relevant protein aggregation. Hence, I am very familiar with tissue culture, transgenic animals as disease models, studies of protein aggregation, and as a former rheumatologist, osteoarthritis.

We greatly appreciate the Reviewer providing such valuable and scholarly input from the perspective of a scientist with deep expertise in the secretory pathway and other diseases of protein misfolding, as well as from rheumatology. Specifically from the perspective of expertise in collagen biology/biochemistry, we hope that our detailed explanations of assays that are possible versus not possible with collagen in this system, the additional context for why our assessment of the modification of procollagen is correlated with folding/secretion rate, and the further analyses added to the paper, now make a convincing case that the improved manuscript is of high significance and is ready for publication.

June 11, 2024

RE: Life Science Alliance Manuscript #LSA-2024-02842-T

Prof. Matthew Shoulders
Massachusetts Institute of Technology
Chemistry
77 Massachusetts Avenue, Building 16-573A
16-573A
Cambridge, MA 2139

Dear Dr. Shoulders,

Thank you for submitting your revised manuscript entitled "ER procollagen storage defect without coupled unfolded protein response drives precocious arthritis". We would be happy to publish your paper in Life Science Alliance pending final revisions necessary to meet our formatting guidelines.

- please carefully consider Reviewer 2's remaining comments
- please be sure that the authorship listing and order is correct
- please upload all figure files as individual ones, including the supplementary figure files; all figure legends should only appear in the main manuscript file
- please add your main, supplementary figure, and table legends to the main manuscript text after the references section and remove them from the figures
- please add ORCID ID for the corresponding author -- you should have received instructions on how to do so
- please add a Summary Blurb/Alternate Abstract and a Category for your manuscript in our system
- please add the Twitter handle of your host institute/organization as well as your own or/and one of the authors in our system
- please add an Author Contributions section to your main manuscript text
- please use the [10 author names et al.] format in your references (i.e., limit the author names to the first 10)
- please add callouts for Figures 5C; 8A-B; S1A-C; S2A-B and Table S1 to your main manuscript text
- please add a Data Availability statement at the end of the Materials and Methods section to include accession information for RNA-seq and mass spec data
- the supplementary methods and reference included in a separate file should be incorporated into the main manuscript file

A. FINAL FILES:

-- Summary blurb (enter in submission system): A short text summarizing in a single sentence the study (max. 200 characters including spaces). This text is used in conjunction with the titles of papers, hence should be informative and complementary to the title. It should describe the context and significance of the findings for a general readership; it should be written in the

present tense and refer to the work in the third person. Author names should not be mentioned.

B. MANUSCRIPT ORGANIZATION AND FORMATTING:

Sincerely,

Reviewer #1 (Comments to the Authors (Required)):

I am satisfied with the modifications that have been implemented. I think the paper reads better following the changes and response to reviewers. Ready to publish in my opinion.

Reviewer #2 (Comments to the Authors (Required)):

The authors have responded appropriately to my prior comments and explained why some of my experimental suggestions were not feasible in the context of chondronoid processing. My major issue is still one of style, i.e. the mixing of discussion and results in what should be the data section. I find this distracts the reader from the flow of the data and would be expressed better in the discussion sections which could refer back to the data. I have a few other points for the authors to consider, but do not have to be addressed in the current paper, but some do.

Line 84: Do you want to put a reference to mouse models here?

Line 132: Is this where cell number/cell death/ apoptosis data (Fig 8) should be inserted as possibly accounting for small size of homozygote chondronoids?

Line 158: "A healthy cartilage matrix.....ossification". This sentence is discussion, not results.

Lines 172-76: Inclusion of these references is helpful to the reader. Was the pattern of post-translational modification the same and just more extensive or was it biochemically qualitatively different? If it were one or the other would it be significant?

Lines 212-216: "These observations.....the ER." This comment is discussion, not results.

Lines 223-225: "Consistent with.....pathologic joints." This sentence is discussion, not results.

Lines 230-237: Not really results but are appropriate here since they lead to the results of your next series of experiments, i.e. lines 239-47.

Lines 273-279: Discussion, not results. Also lines 282 (from "The implications...") -305 is largely interpretation of the data and giving it context rather than presentation.

Lines 310-347: Here the integration of the data with interpretation is stylistically justified.

Lines 349-353: Discussion

Lines 371-380: Here you might wonder what chondronoids from these mice might have looked like. Do you think the in vivo organismal environment of the mouse amplifies the effects of the mutant protein phenotype? Does that environment differ from that of humans?

Lines 415-423: Is it possible that the mutant collagen structure precludes binding of any of the known elements of the UPR to the protein at a physiologically realistic rate so that enhancing the UPR quantitatively would not have a significant effect although this is not suggested by your table 1?

Lines 455-459: I like this.

Lines 460-462: Another possibility for the lack of phenotypes in heterozygote mouse-human models is that the mouse version of the human protein in question is more stable and that the heteromers take on the stability characteristics of the mouse. Not relevant to your work but clearly true is several protein systems.

Fig 5C: Are the apparent peri-nuclear localization differences among WT, heterozygous and homozygous cells meaningful?

Figure S5: transcriptionally, it almost looks like the heterozygote and homozygote chondronoids are two different cell types when compared to the WT by GO analysis. Do you think you should include a direct comparison of the transcriptional analysis of +/- and -/- chondronoids.

Figure S6: It appears that protein degradation, non-proteasomal ubiquitin independent, is lower in the homozygous mutant chondronoids, and you have not mentioned defective ERAD. Perhaps you should?

Referee Cross Comments: I found reviewer 1's comments cogent and relevant and from someone who is much more collagen-savvy than I am. I agree with their pointing out that time might be an important variable in a comparison between the in vivo models and the chondronoids, as well as possibly influencing the transcriptional phenotypes, to which the authors responded appropriately.

Response to Reviewer 2

The authors have responded appropriately to my prior comments and explained why some of my experimental suggestions were not feasible in the context of chondronoid processing. My major issue is still one of style, i.e. the mixing of discussion and results in what should be the data section. I find this distracts the reader from the flow of the data and would be expressed better in the discussion sections which could refer back to the data. I have a few other points for the authors to consider, but do not have to be addressed in the current paper, but some do.

We appreciate the Reviewer's careful assessment of our work. We have made modifications to improve this point, while also trying to retain key places where we think interpreting data is necessary to help readers follow the narrative.

Line 84: Do you want to put a reference to mouse models here?

Thanks for this point, we do prefer to keep the reference initially where it is in the manuscript to retain the narrative.

Line 132: Is this where cell number/cell death/ apoptosis data (Fig 8) should be inserted as possibly accounting for small size of homozygote chondronoids?

We have considered this suggestion but feel that the RNA-sequencing data showing the lack of transcriptional stress response is important context to precede the apoptosis assay.

Line 158: "A healthy cartilage matrix.....ossification". This sentence is discussion, not results.

Lines 212-216: "These observations.....the ER." This comment is discussion, not results.

Lines 223-225: "Consistent with.....pathologic joints." This sentence is discussion, not results.

Lines 230-237: Not really results but are appropriate here since they lead to the results of your next series of experiments, i.e. lines 239-47.

Lines 273-279: Discussion, not results. Also lines 282 (from "The implications...") -305 is largely interpretation of the data and giving it context rather than presentation.

Lines 310-347: Here the integration of the data with interpretation is stylistically justified.

Lines 349-353: Discussion

Lines 455-459: I like this.

We have carefully considered the proposed edits. For these however, we think the brief interpretation of data is important to help readers follow the experimental narrative.

Lines 172-76: Inclusion of these references is helpful to the reader. Was the pattern of post-translational modification the same and just more extensive or was it biochemically qualitatively different? If it were one or the other would it be significant?

We have not characterized the nature of the modifications in depth, as that would be a major experimental effort itself requiring collaboration with an expert mass spectrometry lab and there will be substantial technical challenges. This is a great suggestion for future work however!

Lines 371-380: Here you might wonder what chondronoids from these mice might have looked like. Do you think the in vivo organismal environment of the mouse amplifies the effects of the mutant protein phenotype? Does that environment differ from that of humans?

We agree that this is an interesting direction that could be followed up in future work.

Lines 415-423: Is it possible that the mutant collagen structure precludes binding of any of the known elements

of the UPR to the protein at a physiologically realistic rate so that enhancing the UPR quantitatively would not have a significant effect although this is not suggested by your table 1?

Lines 460-462: Another possibility for the lack of phenotypes in heterozygote mouse-human models is that the mouse version of the human protein in question is more stable and that the heteromers take on the stability characteristics of the mouse. Not relevant to your work but clearly true is several protein systems.

While possible, we do not have data that address these hypotheses and therefore would prefer to avoid speculating.

Fig 5C: Are the apparent peri-nuclear localization differences among WT, heterozygous and homozygous cells meaningful?

We have not found the peri-nuclear localization to be substantially different between genotypes.

Figure S5: transcriptionally, it almost looks like the heterozygote and homozygote chondronoids are two different cell types when compared to the WT by GO analysis. Do you think you should include a direct comparison of the transcriptional analysis of -/+ and -/- chondronoids.

We agree that the heterozygote and homozygote chondronoids are transcriptionally quite distinct. We discuss this point briefly in the discussion section, and have provided all the GSEA comparisons in our supplementary data.

Figure S6: It appears that protein degradation, non-proteasomal ubiquitin independent, is lower in the homozygous mutant chondronoids, and you have not mentioned defective ERAD. Perhaps you should?

Given the high false discovery rate of this particular gene set (FDR of ~10%), we do not believe that it provides enough evidence to speculate extensively regarding defective ERAD. However, this is a great suggestion for future work.

Referee Cross Comments: I found reviewer 1's comments cogent and relevant and from someone who is much more collagen-savvy than I am. I agree with their pointing out that time might be an important variable in a comparison between the in vivo models and the chondronoids, as well as possibly influencing the transcriptional phenotypes, to which the authors responded appropriately.

We are grateful for Reviewer 2's (and Reviewer 1's) assessment and thoughtful input, which have greatly improved the paper throughout the review process.

June 26, 2024

RE: Life Science Alliance Manuscript #LSA-2024-02842-TR

Prof. Matthew Shoulders
Massachusetts Institute of Technology
Chemistry
77 Massachusetts Avenue, Building 16-573A
16-573A
Cambridge, MA 02139

Dear Dr. Shoulders,

Thank you for submitting your Research Article entitled "ER procollagen storage defect without coupled unfolded protein response drives precocious arthritis". It is a pleasure to let you know that your manuscript is now accepted for publication in Life Science Alliance. Congratulations on this interesting work.

DISTRIBUTION OF MATERIALS:

Again, congratulations on a very nice paper. I hope you found the review process to be constructive and are pleased with how the manuscript was handled editorially. We look forward to future exciting submissions from your lab.

Sincerely,
